# Best-of-Both-Worlds Linear Contextual Bandits

**Masahiro Kato**  *mkato-csecon@g.ecc.u-tokyo.ac.jp*
*The University of Tokyo*

**Shinji Ito**  *shinji@mist.i.u-tokyo.ac.jp*
*The University of Tokyo*
*RIKEN AIP*
*NEC Corporation (affiliation upon submission)*

**Reviewed on OpenReview:** *https://openreview.net/forum?id=aIG2RAtNuX&referrer*

## Abstract

This study investigates the problem of $K$-armed linear contextual bandits, an instance of the multi-armed bandit problem, under an adversarial corruption. At each round, a decision-maker observes an independent and identically distributed context and then selects an arm based on the context and past observations. After selecting an arm, the decision-maker incurs a loss corresponding to the selected arm. The decision-maker aims to minimize the cumulative loss over the trial. The goal of this study is to develop a strategy that is effective in both stochastic and adversarial environments, with theoretical guarantees. We first formulate the problem by introducing a novel setting of bandits with adversarial corruption, referred to as the contextual adversarial regime with a self-bounding constraint. We assume linear models for the relationship between the loss and the context. Then, we propose a strategy that extends the `RealLinExp3` by Neu & Olkhovskaya (2020) and the Follow-The-Regularized-Leader (FTRL). The regret of our proposed algorithm is shown to be upper-bounded by $O\left(\min\left\{\frac{\log^3(T)}{\Delta_*} + \sqrt{\frac{C\log^3(T)}{\Delta_*}}, \ \sqrt{T}\log^2(T)\right\}\right)$, where $T \in \mathbb{N}$ is the number of rounds, $\Delta_* > 0$ is the constant minimum gap between the best and suboptimal arms for any context, and $C \in [0, T]$ is an adversarial corruption parameter. This regret upper bound implies $O\left(\frac{\log^3(T)}{\Delta_*}\right)$ in a stochastic environment and by $O\left(\sqrt{T}\log^2(T)\right)$ in an adversarial environment. We refer to our strategy as the `Best-of-Both-Worlds (BoBW) RealFTRL`, due to its theoretical guarantees in both stochastic and adversarial regimes.

## 1 Introduction

This study considers minimizing the cumulative regret in the multi-armed bandit (MAB) problem with contextual information. The MAB problem is a formulation of sequential decision-making. In this study, we develop an algorithm that utilizes side information called contextual information. We focus on linear contextual bandits and aim to design an algorithm that performs well in both stochastic and adversarial environments.

In our problem setting of contextual bandits, a decision-maker observes an independent and identically distributed (i.i.d.) context each round, draws an arm accordingly, and incurs a loss associated with the chosen arm. Additionally, we assume linear models between the loss and contexts, which is known as the linear contextual bandit problem. The contextual bandit problem is widely studied in fields such as sequential treatment allocation (Tewari & Murphy, 2017), personalized recommendations (Beygelzimer et al., 2011), and online advertising (Li et al., 2010). Based on these demands, existing studies explore the methods. For example, Abe & Long (1999) studies linear contextual bandits. Li et al. (2021) provides lower bounds. There are numerous other studies in this field (Chen et al., 2020; Ding et al., 2022).

The settings of linear contextual bandits are divided into stochastic, with fixed contextual and loss distributions, and adversarial environments, with fixed contexts but adversarially chosen losses[1]. Most existing studies focus on algorithms for either stochastic (Abe & Long, 1999; Rusmevichientong & Tsitsiklis, 2010; Chu et al., 2011; Abbasi-yadkori et al., 2011; Lattimore & Szepesvari, 2017) or adversarial linear contextual bandits (Neu & Olkhovskaya, 2020).

Thus, optimal algorithms typically differ between the stochastic and adversarial environments. However, a best-of-both-worlds framework exists, aiming for algorithms that are competent in both stochastic and adversarial environments (Bubeck & Slivkins, 2012; Seldin & Slivkins, 2014; Auer & Chiang, 2016; Seldin & Lugosi, 2017; Zimmert & Seldin, 2021; Lee et al., 2021). Building on existing work, we propose a best-of-both-worlds algorithm for stochastic and adversarial linear contextual bandits.

## 1.1 Main Contribution

In Section 2, we first introduce the setting of linear contextual bandits with adversarial corruption by defining the linear contextual adversarial regime with a self-bounding constraint. This setting is a generalization of the *adversarial regime with a self-bounding constraint* proposed by Zimmert & Seldin (2021). Under this regime, we bridge the stochastic and adversarial environments by an adversarial corruption parameter $C \geq 0$, where $C = 0$ corresponds to a stochastic environment and $C = T$ corresponds to an adversarial environment.

Then, in Section 3 inspired by the `RealLinEXP3` proposed by Neu & Olkhovskaya (2020) for adversarial contexts, our algorithm uses the Follow-the-Regularized-Leader (FTRL) approach to adapt well to the stochastic environment. Our algorithm design also follows existing studies in best-of-both-worlds (BoBW) studies, such as Ito et al. (2022). We refer to our algorithm as the `BoBW-RealFTRL`.

In Section 4, we show the following upper bound of the `BoBW-RealFTRL`:

$$
O\left( \min\left\{ \frac{K \log(T)\left(\log(T) + d \log(K)\right) \log(KT)}{\Delta_*} + \sqrt{\frac{CK \log(T)\left(\log(T) + d \log(K)\right) \log(KT)}{\Delta_*}}, \right.\right.
$$
$$
\left.\left. \sqrt{\log(KT) T K \log(T)\left(\log(T) + d \log(K)\right) \log(KT)} \right\}\right)
$$

where $T$ is the number of rounds, $d$ is the dimension of a context, and $K$ is the number of arms, when there exists a constant minimum gap $\Delta_*$ between the conditional expected rewards of the best and suboptimal arms for any context when we consider a stochastic environment. Note that this regret upper bound holds both for stochastic and adversarial environments. As a specific case, under a stochastic environment, the regret upper bound is given as

$$
O\left( \frac{K \log(T)\left(\log(T) + d \log(K)\right) \log(KT)}{\Delta_*} \right).
$$

When there does not exist such a gap $\Delta_*$, we show that the regret upper bound is given as

$$
O\left( \sqrt{\log(KT) T K \log(T)\left(\log(T) + d \log(K)\right) \log(KT)} \right).
$$

Note that our regret upper bound is $O\left( \min\left\{ \frac{\log^3(T)}{\Delta_*} + \sqrt{\frac{C \log^3(T)}{\Delta_*}}, \ \sqrt{T} \log^2(T) \right\} \right)$ when focusing on the order with respect to $T$. Furthermore, in a stochastic environment, the regret is upper bounded by $O\left( \frac{\log^3(T)}{\Delta_*} \right)$, and in an adversarial environment, the regret is upper bound by $O\left( \sqrt{T} \log^2(T) \right)$.

---

[1]We can define adversarial linear contextual bandits in different ways. For example, there are studies that consider contextual bandits with adversarial contexts and fixed losses (Chu et al., 2011; Abbasi-yadkori et al., 2011). On the other hand, several studies address contextual bandits with adversarial contexts and adversarial losses (Kanade & Steinke, 2014; Hazan et al., 2016). This study only focuses on contextual bandits with i.i.d. contexts and adversarial losses, which have been studied by Rakhlin & Sridharan (2016) and Syrgkanis et al. (2016).

In summary, we contribute to the field of linear contextual bandits by providing a BoBW algorithm. Our proposed algorithm and theoretical analysis draw partial inspiration from Neu & Olkhovskaya (2020) and Ito et al. (2022). However, we introduce several novel techniques to adapt these methodologies to our setting, such as specifying adaptation weights based on an estimation algorithm for the covariance matrix of contexts. It is important to note that our primary contribution lies not in theoretical advancements but in the proposition of a novel algorithm tailored to a unique setting. Our study significantly enriches the domains of linear contextual bandits and BoBW algorithms.

## 1.2 Related Work

In adversarial bandits, the `RealLinExp3`, the algorithm proposed by Neu & Olkhovskaya (2020), yields $O\left(\log(T)\sqrt{KdT}\right)$. In Table 1, we compare our regret upper bounds with the upper bounds of Neu & Olkhovskaya (2020).

Regret upper bounds in a stochastic setting are categorized into problem-dependent and problem-independent upper bounds, where the former utilizes some distributional information, such as the gap parameter $\Delta_*$, to bound the regret, while the latter does not. Additionally, problem-dependent regret upper bounds in the stochastic bandits depend on the margin condition characterized by a parameter $\alpha \in [0, +\infty]$ (for the detailed definition, see Remark 1). Our case with $\Delta_*$ corresponds to a case with $\alpha = +\infty$. Note that in the adversarial bandits, the margin condition usually does not affect the upper bounds. Dani et al. (2008) proposes the `ConfidenceBAll`, and Abbasi-yadkori et al. (2011) proposes `OFUL`. They both present upper bound with and without the assumption of the existence of $\Delta_*$[2]. As mentioned above, the regret upper bound under the assumption of the existence of $\Delta_*$ corresponds to a case with $\alpha = +\infty$ in the margin condition. In contrast, Goldenshluger & Zeevi (2013), Wang et al. (2018), and Bastani & Bayati (2020) propose algorithm in a case with $\alpha = 1$. Furthermore, Li et al. (2021) propose the $\ell_1$-`ConfidenceBall` based algorithm whose upper bound tightly depends on unknown $\alpha$. Note that their setting of linear contextual bandits slightly differs from the one presented in Neu & Olkhovskaya (2020). While Neu & Olkhovskaya (2020) considers a scenario where contexts remain constant across arms but the coefficients of linear models vary, other studies typically adopt a setting where each arm has its specific context, but the coefficients are the same. In this latter context, Abbasi-yadkori et al. (2011) demonstrates that their `OFUL` algorithm achieves an $O\left(d\log(T)/\Delta_*\right)$ regret upper bound. It is important to note that due to these differing settings, we do not compare compare their regret bounds to ours directly and omit it from Table 1.

There are several related studies for linear contextual bandits with adversarial corruption, including Lykouris & Vassilvtiskii (2018), Gupta et al. (2019), Zhao et al. (2021) and He et al. (2022). Lykouris & Vassilvtiskii (2018), Gupta et al. (2019), and Zhao et al. (2021) consider other corruption frameworks characterized by a constant $\widetilde{C} \in [0, T]$, which is different but related to our linear contextual adversarial regime with a self-bounding constraint. He et al. (2022) uses another constant $\widetilde{C}^\dagger \in [0, T]$ different but closely related to $\widetilde{C}$. For the detailed definitions, see Remark 2. The essential difference between our and their settings is the existence of the gap $\Delta_*$. Furthermore, while our regret upper bound achieves the polylogarithmic order, those studies show roughly $\sqrt{T}$-order regret upper bounds. He et al. (2022) presents $\widetilde{O}\left(d\sqrt{K} + d\widetilde{C}^\dagger\right)$ regret under an adversarial corruption characterized by a constant $\widetilde{C}^\dagger > 0$.

The use of the FTRL approach for adversarial linear bandits is also independently explored by Liu et al. (2023) to relax the assumption used in Neu & Olkhovskaya (2020). In addition to the difference in contributions, while our algorithm utilizes the Shannon entropy in the regularization of the FTRL, Liu et al. (2023) employs the log-determinant barrier. We expect that combining these two methods will yield a BoBW algorithm with relaxed assumptions, and it is future work.

To establish our BoBW regret bounds, we utilize the self-bounding technique (Zimmert & Seldin, 2021; Wei & Luo, 2018), which yields poly-logarithmic regret bounds in stochastic environments. This is achieved by integrating regret upper bounds that are contingent on the arm-selection distributions $q_t$, and a lower

---

[2]Regret upper bounds with the assumption of the existence of $\Delta_*$ are called problem-dependent.

Table 1: Comparison of the regret. We compared the regret upper bound of our proposed `BoBW-RealFTRL` with the `RealLinExp3` (Neu & Olkhovskaya, 2020).

| | Regret | Adversarial/Stochastic |
|---|---|---|
| Ours (`BoBW-RealFTRL`) | $O\left(\min\left\{\frac{D}{\Delta_*} + \sqrt{\frac{CD}{\Delta_*}}, \sqrt{\log(KT)TD}\right\}\right)$ where $D = K\log(T)\left(\log(T) + d\log(K)\right)\log(KT)$ | Both |
| | $O\left(\sqrt{\log(KT)TD}\right)$ | Adversarial |
| | $O\left(\frac{D}{\Delta_*}\right)$ $(C = 0)$ | Stochastic |
| `RealLinExp3` | $O\left(\log(T)\sqrt{KdT}\right)$ | Adversarial |

bound known as self-bound constraints. The $q_t$-dependent regret bounds are obtained using FTRL with a negative-entropy regularizer, which is also referred to as the *exponential weight* method.

Our approach includes an entropy-adaptive update rule for learning rates, originally developed for online learning in feedback graph contexts (Ito et al., 2022). This strategy has been proven effective in providing BoBW guarantees for exponential-weight-based algorithms across various sequential decision-making problems, such as multi-armed bandits (Jin et al., 2023), partial monitoring (Tsuchiya et al., 2023a), linear bandits (Kong et al., 2023), episodic Markov Decision Processes (MDPs) (Dann et al., 2023b), and sparse multi-armed bandits (Tsuchiya et al., 2023b). However, a common limitation of these results, stemming from the negative-entropy regularization, is the additional $\log T$ factors in the regret bounds.

Our obtained regret upper bounds are not $O(\log(T))$, and we assume the uniqueness of the optimal arm. A promising future direction to address these issues could be the exploration of alternative regularizers like Tsallis entropy. Utilizing such regularizers might allow us to tighten the regret upper bound and relax the assumption that the optimal arm is unique. Zimmert & Seldin (2021) proposes the use of Tsallis entropy, and Masoudian & Seldin (2021) enhances this approach, yielding tighter upper bounds with respect to $T$. We anticipate that by employing such regularizers, we could achieve $\log(T)$ or $\log^2(T)$ regret upper bounds, similar to the `OFUL` algorithm in Abbasi-yadkori et al. (2011). We expect that the regret upper bounds will be worse than the one in Abbasi-yadkori et al. (2011) due to the estimation of the inverse covariance matrix. While alternative regularization could be applicable in our context, it may require additional assumptions to derive desired regret upper bounds. Concerning the number of optimal arms, Ito (2021) and Jin et al. (2023) introduce BoBW algorithms for scenarios where the optimal arm is not unique in standard multi-armed bandits. However, the analysis is complex, and extensions to combinatorial semi-bandits or linear bandits, which are arguably simpler than contextual linear bandits, are yet to be seen. Thus, although their methods hold promise for linear contextual bandits, their straightforward application in our setting presents significant challenges.

Independently of us, Kuroki et al. (2024) proposes several FTRL-based BoBW algorithms for linear contextual bandits with adversarial corruptions. One of their proposed algorithms is also a BoBW algorithm based on FTRL with Shannon entropy, which coincides with our proposed algorithm. As another approach, they employ the black-box framework proposed by Dann et al. (2023a). They show that under the black-box framework, we can obtain a BoBW reduction of RealEXP3 with a regret of order $O(\log(T))$ in the stochastic regime. They also present another algorithm by combining the black-box framework with the continuous exponential weights algorithm in Olkhovskaya et al. (2023). While this algorithm incurs a regret of $O(\log^5(T))$ in the stochastic regime, it incurs a regret of $O(\log(T)dK\sqrt{\Lambda^*})$ in the adversarial regime, where $\Lambda^*$ denotes the cumulative second moment of the losses incurred by the algorithm. Although the black-box framework provides a tight $O(\log(T))$ regret regarding $T$, limitations have been reported in practical implementation. As well as Kuroki et al. (2024) mentions, "FTRL with Shannon entropy regularization is a much more practical algorithm."

## 2  Problem Setting

Suppose that there are $T$ rounds and $K$ arms. In each round $t \in [T] \coloneqq \{1, 2, \ldots, T\}$, a decision-maker observes a context $X_t \in \mathcal{X} \subset \mathbb{R}^d$, where $\mathcal{X}$ is a context space. Then, the decision-maker chooses an arm

$A_t \in [K] \coloneqq \{1, 2, \dots, K\}$ based on the context $X_t$ and past observations. Each arm $a \in [K]$ is linked to a loss $\ell_t(a, X_t)$, which depends on $X_t$ and round $t$. After choosing arm $A_t$ in round $t$, the decision-maker incurs the loss $\ell_t(A_t, X_t)$. Our goal is to minimize the cumulative loss $\sum_{t=1}^{T} \ell_t(A_t, X_t)$. We introduce the setting in more detail in the following part.

**Contextual distribution.** Let a distribution of $X_t$ be $\mathcal{D}$, which is invariant across $t \in [T]$. We also assume that $\mathcal{D}$ is known to the decision-maker.

**Assumption 2.1** (Contextual distribution). *The context $X_t$ is an i.i.d. random variable, whose distribution $\mathcal{D}$ is **known** to the decision-maker, and the covariance matrix $\Sigma = \mathbb{E}\left[X_t X_t^\top\right]$ is positive definite with its smallest eigenvalue $\lambda_{\min} > 0$. We also assume that $\lambda_{\min}$ is* known.

## 2.1 Linear Contextual Bandits

This study assumes linear models between $\ell_t(a, X_t)$ and $X_t$ as follows.

**Assumption 2.2** (Linear models). *For each $\ell_t(a, X_t)$, the following holds:*

$$\ell_t(a, X_t) = X_t^\top \theta_t(a) + \varepsilon_t(a),$$

*where $\theta_t(a) \in \Theta$ is a d-dimensional parameter with a parameter space $\Theta \subset \mathbb{R}^d$, and $\varepsilon_t(a)$ is the error term independent of the sequence $\{X_t\}_{t \in [T]}$.*

In each round $t \in [T]$, a context is sampled from $\mathcal{D}$ and the environment chooses $\{\theta_t(a)\}_{a \in [K]}$ based on the past observations $\mathcal{F}_{t-1} = (X_1, A_1, \ell_1(A_1, X_1), X_2, \dots, X_{t-1}, A_{t-1}, \ell_{t-1}(A_{t-1}, X_{t-1}))$.

For linear models and variables, we make the following assumptions.

**Assumption 2.3** (Bounded variables). *We assume the following:*

1. *There exists an universal constant $C_{\mathcal{X}} > 0$ such that for each $x \in \mathcal{X}$, $\|x\|_2 \leq C_{\mathcal{X}}$ holds.*

2. *There exists an universal constant $C_{\Theta} > 0$ such that for each $\theta \in \Theta$, $\|\theta\|_2 \leq C_{\Theta}$ holds.*

3. *There exists an universal constant $C_{\mathcal{E}} > 0$ such that $|\varepsilon_t(a)| \leq C_{\mathcal{E}}$ holds.*

Under this assumption, there exists $C_\ell \coloneqq C(C_{\mathcal{X}}, C_{\Theta}, C_{\mathcal{E}})$ such that for all $\ell_t(a, x)$, the following holds for each $a \in [K]$ and $x \in \mathcal{X}$:

$$|\ell_t(a, x)| \leq C_\ell.$$

For simplicity, we assume $C_\ell = 1$.

We consider a general framework where $\{\theta_t(a)\}_{a \in [K]}$ is generated in both stochastic and adversarial ways. See Section 2.5 for details.

## 2.2 Policy

We refer to a function that determines the arm draw as a policy. Let $\Pi$ be a set of all possible policies $\pi : \mathcal{X} \to \mathcal{P} \coloneqq \left\{ u = (u_1 \ u_2 \ \dots \ u_K)^\top \in [0, 1]^K \mid \sum_{k=1}^{K} u_k = 1 \right\}$. Let $\pi(a \mid x)$ be the $a$-th element of $\pi(x)$. The goal of the decision-maker is to minimize the cumulative loss $\sum_{t=1}^{T} \ell_t(A_t, X_t)$ incurred through $T$ rounds by learning a policy $\pi \in \Pi$.

## 2.3 Data-Generating Process

In each round of a trial, the decision-maker first observes a context and then chooses an action based on the context and past observations obtained until the round. Specifically, we consider sequential decision-making with the following steps in each round $t \in [T]$:

1. The environment decides $\{\theta_t(a)\}_{a \in [K]}$ based on $\mathcal{F}_{t-1}$.

2. The decision-maker observes the context $X_t$, which is generated from a known distribution $\mathcal{D}$.

3. Based on the context $X_t$, the decision-maker chooses a policy $\pi_t(X_t) \in \mathcal{P}$.

4. The decision-maker chooses action $A_t \in [K]$ with probability $\pi_t(a \mid X_t)$.

5. The decision-maker incurs the loss $\ell_t(A_t, X_t)$.

The goal of the decision-maker is to choose actions in a way that the total loss is as small as possible.

## 2.4 Regret

This section provides the definition of the regret, a relative measure of the cumulative loss. We evaluate the performance of the decision or policy of the decision-maker by using regret. Let $\mathcal{R}$ be a set of all possible $\rho : \mathcal{X} \to [K]$. The quality of a decision by the decision-maker is measured by its total expected regret, defined as

$$R_T = \max_{\rho \in \mathcal{R}} \mathbb{E}\left[\sum_{t=1}^{T} \left\{\ell_t(A_t, X_t) - \ell_t(\rho(X_t), X_t)\right\}\right] = \max_{\rho \in \mathcal{R}} \mathbb{E}\left[\sum_{t=1}^{T} \left\langle X_t, \theta_t(A_t) - \theta_t(\rho(X_t))\right\rangle\right],$$

where the expectation is taken over the randomness of policies of the decision-maker, as well as the sequence of random contexts, $\{X_t\}_{t \in [T]}$, and losses, $\{\ell_t(\cdot, X_t)\}_{t \in [T]}$.

Let $X_0$ be an i.i.d. random variable from the distribution of $X_t$. Then, because $X_t$ is an i.i.d. random variable from $\mathcal{D}$, we have

$$\mathbb{E}\left[\sum_{t=1}^{T} \left\langle X_t, \theta_t(\rho(X_t))\right\rangle\right] = \mathbb{E}\left[\sum_{t=1}^{T} \left\langle X_0, \theta_t(\rho(X_0))\right\rangle\right] \geq \mathbb{E}\left[\min_{a \in [K]} \sum_{t=1}^{T} \left\langle X_0, \mathbb{E}\left[\theta_t(a)\right]\right\rangle\right].$$

Based on this inequality, we define an optimal policy $a_T^*$ as

$$a_T^*(x) = \arg\min_{a \in [K]} \sum_{t=1}^{T} \left\langle x, \mathbb{E}\left[\theta_t(a)\right]\right\rangle.$$

Then, we have

$$R_T \leq \mathbb{E}\left[\sum_{t=1}^{T} \left\langle X_t, \theta_t(A_t) - \theta_t(a_T^*(X_t))\right\rangle\right].$$

Neu & Olkhovskaya (2020) refers to $\rho$ as linear-classifier policies, while $\pi_t$ is called stochastic policies. In our study, decision-makers compare their stochastic policies $\pi_t$ to the optimal linear-classifier policy $a^*$ using the regret.

## 2.5 Linear Contextual Adversarial Regime with a Self-Bounding Constraint

Then, we define the framework of a *linear contextual adversarial regime with a self-bounding constraint*, which is a generalization of adversarial and stochastic bandits.

**Definition 2.4** (Linear contextual adversarial regime with a self-bounding constraint). We say that an environment is in a linear contextual adversarial regime with a $(\Delta_*, C, T)$ self-bounding constraint for some $\Delta_*, C > 0$ if $R_T$ is lower bounded as

$$R_T \geq \Delta_* \cdot \mathbb{E}\left[\sum_{t=1}^{T} \left(1 - \pi_t(a_T^*(X_0) \mid X_0)\right)\right] - C.$$

The contextual adversarial regime with a self-bounding constraint includes several important settings. Among them, we raise linear contextual bandits in stochastic bandits and adversarial bandits below. Both correspond to cases with $C = 0$ (linear contextual adversarial regime with a $(\Delta_*, 0, T)$ self-bounding constraint) and $C = 2T$ (linear contextual adversarial regime with a $(\Delta_*, 2T, T)$ self-bounding constraint), respectively. The parameter $C$ determines the degree of an adversarial corruption: as $C$ is larger, the adversarial corruption is also larger.

**Example 1** (Linear contextual bandits in stochastic bandits.). We consider a linear contextual adversarial regime with a $(\Delta_*, 0, T)$ self-bounding constraint, which corresponds to a stochastic environment. In stochastic bandits, the bandit models are fixed; that is, $(X_t, \ell_t(1, X_t), \ldots, \ell_t(K, X_t))$ are generated from a fixed distribution $P_0$. Let $\theta_1(a) = \cdots \theta_T(a) = \theta_0(a)$. Note that when considering stochastic bandits, we have $\mathbb{E}[\theta_t(a)] = \theta_0(a)$ and

$$a_T^*(x) = \underset{a \in [K]}{\arg\min} \sum_{t=1}^T \left\langle x, \mathbb{E}[\theta_t(a)] \right\rangle = \underset{a \in [K]}{\arg\min} \left\langle x, \theta_0(a) \right\rangle \qquad \forall x \in \mathcal{X}.$$

Let $a_T^*(x)$ be $a_0^*(x)$.

In this setting, we assume that for each $P_0$, there exist positive constraints $\Delta_*$ such that for all $x \in \mathcal{X}$,

$$\min_{b \neq a_0^*(x)} \left\langle x, \theta_0(b) \right\rangle - \left\langle x, \theta_0(a_0^*(x)) \right\rangle \geq \Delta_*. \tag{1}$$

Then, the regret can be lower bounded as $R_T \geq \Delta_* \cdot \mathbb{E}\left[\sum_{t=1}^T \left(1 - \pi_t(a_0^*(X_0) \mid X_0)\right)\right]$ (See Appendix A).

**Example 2** (Linear contextual bandits in adversarial bandits). We consider a linear contextual adversarial regime with a $(\Delta_*, 2T, T)$ self-bounding constraint, which corresponds to a stochastic environment In adversarial bandits, we do not assume any data-generating process for the $\ell_t(a, X_t)$, and the loss is decided to increase the regret based on the past observations $\mathcal{F}_{t-1}$.

**Remark 1** (Margin conditions). In linear contextual bandits, we often employ the *margin condition* to characterize the difficulty of the problem instance. The margin condition is defined as follows (Li et al., 2021): there exist $\Delta_*$, $C_1$, $a^*$, and $\alpha \in [0, +\infty]$, such that for $h \in \left[C_1\sqrt{\frac{\log(d)}{T}}, \Delta_*\right]$,

$$\mathbb{P}\left(\left\langle X_t, \theta_t(a^*) \right\rangle \geq \min_{b \neq a^*} \left\langle X_t, \theta_t(b) \right\rangle - h\right) \leq \frac{1}{2}\left(\frac{h}{\Delta_*}\right)^\alpha.$$

Our definition of the linear contextual adversarial regime with a self-bounding constraint corresponds to a case with $\alpha = \infty$. In existing studies of linear contextual bandits under an adversarial environment, the margin condition with $\alpha = 0$ is assumed in Abbasi-yadkori et al. (2011), which demonstrates $O(\sqrt{T})$-regret. On the other hand, the margin condition with $\alpha \in (0, 1]$ is considered in works by Goldenshluger & Zeevi (2009; 2013), Wang et al. (2018), and Bastani & Bayati (2020). Goldenshluger & Zeevi (2009) establishes $O\left(T^{(1-\alpha)/2}\right)$-regret for $\alpha \in (0, 1)$, while Goldenshluger & Zeevi (2013) proves $O\left(d^3\log(T)\right)$-regret for $\alpha = 1$. Furthermore, Abbasi-yadkori et al. (2011) explores a scenario with $\alpha = \infty$ and demonstrates $O(\log(T)/\Delta_*)$-regret. Li et al. (2021) presents a general lower bound that is applicable for various $\alpha$. Extending our results to encompass more general values of $\alpha$ constitutes future work. We anticipate that an ideal BoBW algorithm would perform comparably to the existing studies, such as those by Abbasi-yadkori et al. (2011) and Goldenshluger & Zeevi (2009; 2013), in a stochastic environment.

**Remark 2** (Linear contextual bandits with corruption in existing studies). Lykouris & Vassilvtiskii (2018), Gupta et al. (2019), Zhao et al. (2021), and He et al. (2022) propose another definition of linear contextual contextual bandits with corruption. In their work, instead of our defined $\ell_t(a, X_t)$, they define a loss as

$$\widetilde{\ell}_t(a, X_t) = \ell_t(a, X_t) + \widetilde{c}_t(a),$$

where $\widetilde{c}_t(a)$ is an adversarial corruption term. For simplicity, let $\widetilde{c}_t(a) \in [-1, 1]$. In Zhao et al. (2021), the degree of the corruption is determined by $\widetilde{C} \in [0, T]$ defined as $\widetilde{C} = \sum_{t=1}^T \max_{a \in [K]} |c_t(a)|$. In Lykouris &

Vassilvtiskii (2018), Gupta et al. (2019), and He et al. (2022), the corruption level is determined by another parameter $\widetilde{C}^\dagger \in [0, T]$ defined as $\sum_{t=1}^{T} |c_t(A_t)|$. Here, $\widetilde{C} \geq \widetilde{C}^\dagger$ holds. Note that the adversarial corruption depends on $A_t$ in Zhao et al. (2021), while the adversarial corruption is determined irrelevant to $A_t$ in Lykouris & Vassilvtiskii (2018), Gupta et al. (2019), and He et al. (2022). Unlike ours, they do not assume the existence of $\Delta_*$ defined in equation 1. In this sense, our results and their results are complementary.

## 3 Algorithm

This section provides an algorithm for our defined problem. Our proposed algorithm is a generalization of the `RealLinEXP3` algorithm proposed by Neu & Olkhovskaya (2020). We extend the method by employing the Follow-The-Regularized-Leader (FTRL) approach with round-varying arm-drawing probabilities. Our design of the algorithm is also motivated by existing studies about Best-of-Both-Worlds (BoBW) algorithms in different Multi-Armed Bandit (MAB) problems, such as Ito et al. (2022).

In our setting, we first observe a context and then draw an arm based on that context. We consider stochastically drawing an arm. Therefore, in designing the algorithm, our interest lies in appropriately defining the arm-drawing probability. In the FTRL approach, we define this probability by utilizing an unbiased estimator of the loss function.

We refer to our proposed algorithm as `BoBW-RealFTRL` because it modifies the `RealLinEXP3` for a best-of-both-worlds algorithm using the FTRL framework. The pseudo-code is shown in Algorithm 1. In the following part, we explain the algorithm.

**Unbiased loss estimator.** For each $a \in [K]$, let us define an estimator of regression parameters as

$$\widehat{\theta}_t(a) := \Sigma_{t,a}^\dagger \mathbb{1}[A_t = a] X_t \ell_t(A_t, X_t),$$

where $\Sigma_{t,a}^\dagger$ is an estimator of $\Sigma_t^{-1} := \mathbb{E}\left[\mathbb{1}[A_t = a] X_t^\top X_t \mid \mathcal{F}_{t-1}\right]^{-1}$. Then, the loss can be estimated as

$$\widehat{\ell}_t(a, x) = \left\langle x, \widehat{\theta}_t(a) \right\rangle.$$

In analysis of adversarial bandits, the bias of $\widehat{\ell}_t(a, x)$ plays an important role. If $\Sigma_{t,a}^\dagger = \mathbb{E}\left[\mathbb{1}[A_t = a] X_t^\top X_t \mid \mathcal{F}_{t-1}\right]^{-1}$, then this loss estimator is unbiased for $x^\top \theta_0(a)$ because

$$\mathbb{E}\left[\widehat{\ell}_t(a, x) \mid \mathcal{F}_{t-1}\right] = x^\top \mathbb{E}\left[\widehat{\theta}_t(a) \mid \mathcal{F}_{t-1}\right] = x^\top \Sigma_{t,a}^\dagger \mathbb{E}\left[\mathbb{1}[A_t = a] X_t \ell_t(A_t, X_t) \mid \mathcal{F}_{t-1}\right]$$

$$= x^\top \Sigma_{t,a}^\dagger \mathbb{E}\left[\mathbb{1}[A_t = a] X_t \left\{X_t^\top \theta_0(a) + \varepsilon_t(a)\right\} \mid \mathcal{F}_{t-1}\right] = x^\top \theta_0(a).$$

Note that in our algorithm, $\Sigma_{t,a}^\dagger$ is just an estimator of $\Sigma_t^{-1}$, and $\Sigma_{t,a}^\dagger = \Sigma_t^{-1}$ does not hold in general. Therefore, $\widehat{\ell}_t(a, x)$ is not unbiased. However, we can show that the bias can be ignored because it is sufficiently small to evaluate the regret in depth. We also define a vector of loss estimators as $\widehat{\ell}_t(x) = \left(\widehat{\ell}_t(1, x) \ \ \widehat{\ell}_t(2, x) \ \ \cdots \ \ \widehat{\ell}_t(K, x)\right)^\top$.

**Estimation of $\Sigma_t^{-1}$.** Our remaining task is to estimate $\Sigma_t^{-1}$. The difficulty of this task stems from the dependency on $A_t$, which varies across rounds. To address this issue, we employ the `Matrix Geometric Resampling` (MGR) proposed by Neu & Olkhovskaya (2020). The MGR assumes that we have access to the distribution $\mathcal{D}$ of $X_t$ and estimates $\Sigma_t^{-1}$ by using simulations. We introduce the algorithm in Algorithm 2.

In Algorithm 2, we define $W_{k,j,a}$ so that $\mathbb{E}[W_{k,j,a} \mid \mathcal{F}_{t-1}] = \Sigma_{t,a}$ holds for each $k \in \{1, 2, \ldots, M_t\}$ and $j \in \{1, 2, \ldots, k\}$. Let us define $V_{k,a} = \prod_{j=1}^{k}(I - \delta W_{k,j,a})$. Here, from the independence of the context $W_{k,j,a}$ from each other, we also have

$$\mathbb{E}[V_{k,a} \mid \mathcal{F}_{t-1}] = \mathbb{E}\left[\prod_{j=1}^{k}\left(I - \delta W_{k,j,a}\right) \mid \mathcal{F}_{t-1}\right] = \prod_{j=1}^{k}\left(I - \delta \mathbb{E}\left[W_{k,j,a} \mid \mathcal{F}_{t-1}\right]\right) = (I - \delta \Sigma_{t,a})^k.$$

Therefore, $\widehat{\Sigma}_{t,a}^{\dagger} = \delta I + \delta \sum_{k=1}^{M_t} V_{k,a}$ works as a good estimator of $\Sigma_{t,a}^{-1}$ on expectations when $M_t = \infty$ because

$$\mathbb{E}\left[\widehat{\Sigma}_{t,a}^{\dagger} \mid \mathcal{F}_{t-1}\right] = \delta I + \delta \sum_{k=1}^{\infty}(I - \delta\Sigma_{t,a})^k = \delta \sum_{k=0}^{\infty}(I - \delta\Sigma_{t,a})^k = \delta(\delta\Sigma_{t,a})^{-1} = \Sigma_{t,a}^{-1}.$$

holds, where we used $\sum_{k=0}^{\infty} A^k = (I - A)^{-1}$ for a diagonalizable square matrix $A$ whose all eigenvalues are greater than $-1$ and less than 1. In implementation, $M_t$ is usually finite, which causes an estimation error. Therefore, Neu & Olkhovskaya (2020) performs estimation error analysis of $\Sigma_{t,a}^{-1}$ with finite $M_t$ in Lemma B.5.

**Our proposed algorithm: BoBW-RealFTRL.** Then, we define our policy, called the BoBW-RealFTRL, as

$$\pi_t(X_t) \coloneqq (1 - \gamma_t)q_t(X_t) + \frac{\gamma_t}{K}\iota, \tag{2}$$

where $\iota$ is a $K$-dimensional vector $\iota = (1\ 1\ \cdots\ 1)^{\top}$,

$$q_t(x) \in \underset{q \in \Pi}{\arg\min}\left\{\sum_{s=1}^{t-1}\left\langle \widehat{\ell}_s(x), q(x)\right\rangle + \psi_t(q(x))\right\} \quad \text{for } t \geq 2, \quad q_1(x) \coloneqq (1/K\ 1/K\ \cdots\ 1/K)^{\top},$$

$$\psi_t(q(x)) \coloneqq -\beta_t H(q(x)), \quad H(q(x)) \coloneqq \sum_{a \in [K]} q(a \mid x)\log\left(\frac{1}{q(a \mid x)}\right), \tag{3}$$

$$\beta_{t+1} \coloneqq \beta_t + \frac{\beta_1}{\sqrt{1 + \left(\log(K)\right)^{-1}\sum_{s=1}^{t} H\left(q_s(X_s)\right)}}, \quad \beta_1 \coloneqq C_\Theta C_{\mathcal{X}}^2 \sqrt{K\log(T)\left(\frac{\log(T)}{\delta\lambda_{\min}\sqrt{\log(K)}} + d\right)}, \tag{4}$$

$$\widetilde{\gamma}_t \coloneqq \frac{K}{2\delta\lambda_{\min}\beta_t}\log(T) = \frac{C_\Theta C_{\mathcal{X}}^2 K}{\lambda_{\min}\beta_t}\log(T), \quad \gamma_t = \min\{1, \widetilde{\gamma}_t\}, \tag{5}$$

$$M_t \coloneqq (2\beta_t - 1), \quad \delta \coloneqq \frac{1}{2C_\Theta C_{\mathcal{X}}^2}.$$

Note that an FTRL algorithm with the Shannon entropy is a generalization of the exponential weighting algorithms. Indeed, $q_t(x)$ has an analytical solution given as

$$q_t(x) = \frac{\exp\left(-\sum_{s=1}^{t-1}\widehat{\ell}_s(a, x)/\beta_t\right)}{\sum_{b \in [K]}\exp\left(-\sum_{s=1}^{t-1}\widehat{\ell}_s(b, x)/\beta_t\right)}.$$

The parameters $\beta_t$ and $\gamma_t$ adaptively adjust the regularization and exploration based on the past observations up to the $t$-th round. Appropriately chosen fixed parameters for these can lead to the RealLinEXP3 algorithm proposed by (Neu & Olkhovskaya, 2020), which is tailored for adversarial environments but does not achieve logarithmic regret upper bounds. Therefore, our algorithm extends RealLinEXP3 and FTRL. In the realm of BoBW algorithms, FTRL-based methods are commonly utilized, linking our approach to existing literature. It is crucial to note the balance among $\beta_t$, $\gamma_t$, and $M_t$ (a parameter in the MGR algorithm) is vital for ensuring desirable regret upper bounds. For instance, the exploration rate $\gamma_t$ influences $\Sigma_t^{-1}$, and $M_t$ is consequently affected as it is used in estimating $\Sigma_t^{-1}$. Since $M_t$ also impacts the regret upper bound, parameter selection is meticulously conducted to tighten the bound considering the interplay among $\beta_t$, $\gamma_t$, and $M_t$.

## 4 Regret Analysis

This section provides upper bounds for the regret of our proposed BoBW-RealFTRL algorithm.

For notational simplicity, let us denote $a_T^*$ by $a^*$. To derive upper bounds, we define the following quantities:

$$Q(a^* \mid x) = \sum_{t=1}^{T}\left\{1 - q_t\left(a^*(x) \mid x\right)\right\}, \qquad \overline{Q}(a^*) = \mathbb{E}\left[Q(a^* \mid X_0)\right].$$

---

**Algorithm 1** `BoBW-RealFTRL`.

---

**Parameter:** Learning rate $\beta_1, \beta_2, \ldots, \beta_T > 0$ and exploration parameter $\gamma_t \in (0, 1)$, which depend on $\lambda_{\min}, C_\Theta, C_\mathcal{X}, K, d$, and $T$.
**Initialization:** Set $\theta_0(a) = 0$ for all $a \in [K]$.
**for** $t = 1, \ldots, T$ **do**
    Observe $X_t$.
    Draw $A_t \in [K]$ following the policy $\pi_t(X_t) := (1 - \gamma_t)q_t(X_t) + \frac{\gamma_t}{K}\iota$ defined in equation 2.
    Observe the loss $\ell_t(A_t, X_t)$.
    Compute $\widehat{\theta}_t(a)$ for all $a \in [K]$.
**end for**

---

**Algorithm 2** `Matrix Geometric Resampling` (Neu & Olkhovskaya, 2020).

---

**Input:** Context distribution $\mathcal{D}$, policy $\pi_t$, action $a \in [K]$.
**for** $k = 1, \ldots, M_t$ **do**
    **for** $j = 1, \ldots, k$ **do**
        Draw $X_{k,j} \sim \mathcal{D}$ and $V_{k,j} \sim \pi_t(\cdot \mid X_{k,j})$.
        Compute $W_{k,j,a} = \mathbb{1}[V_{k,j} = a]X_{k,j}X_{k,j}^\top$.
    **end for**
    Compute $V_{k,a} = \prod_{j=1}^{k}(I - \delta W_{k,j,a})$.
**end for**
**Return:** $\widehat{\Sigma}_{t,a}^\dagger = \delta I + \delta \sum_{k=1}^{M_t} V_{k,a}$.

---

Then, we show the following upper bound, which holds for general cases such as adversarial and stochastic environments. We show the proof in Sections B.1 and B.2.

**Theorem 4.1** (General regret bounds). *If the environment generates losses under the contextual adversarial regime with a self-bounding constraint (Definition 2.4), the* `BoBW-RealFTRL` *with* $\widehat{\Sigma}_{t,a}^\dagger$ *incurs the total regret*

$$R_T = O\left(\left(\frac{K\log(T)}{\beta_1}\left(\frac{\log(T)}{\delta\lambda_{\min}\sqrt{\log(K)}} + d\right) + \beta_1\sqrt{\log(K)}\right)\sqrt{\log(KT)}\max\left\{\overline{Q}^{1/2}(a^*), 1\right\}\right).$$

For each situation, such as adversarial environments and linear contextual adversarial regimes with a self-bounding constraint, we derive a specific upper bound.

First, from $\overline{Q}(a^*) \leq T$, the following regret bound holds without any assumptions on the loss; that is, it holds for an adversarial environment.

**Corollary 4.2.** *Assume the same conditions in Theorem 4.1. Then, under an adversarial environment, the regret satisfies*

$$R_T = O\left(\left(\frac{K\log(T)}{\beta_1}\left(\frac{\log(T)}{\delta\lambda_{\min}\sqrt{\log(K)}} + d\right) + \beta_1\sqrt{\log(K)}\right)\sqrt{\log(KT)}\sqrt{T}\right);$$

*that is, from* $\beta_1 = C_\Theta C_\mathcal{X}^2\sqrt{K\log(T)\left(\frac{\log(T)}{\delta\lambda_{\min}\sqrt{\log(K)}} + d\right)}$,

$$R_T = O\left(\log(KT)\sqrt{K\log(K)T\log(T)\left(\frac{\log(T)}{\delta\lambda_{\min}\sqrt{\log(K)}} + d\right)}\right)$$

*holds.*

Furthermore, we derive a regret bound under the linear contextual adversarial regime with a self-bounding constraint.

**Corollary 4.3** (Regret bounds under the linear contextual adversarial regime with a self-bounding constraint). *Suppose that the same conditions in Theorem 4.1 hold. Then, under the contextual adversarial regime with self-bounding constraints, the regret satisfies*

$$R_T = O\Bigg(\bigg\{\frac{K\log(T)}{\beta_1}\left(\frac{\log(T)}{\delta\lambda_{\min}\sqrt{\log(K)}}+d\right)+\beta_1\sqrt{\log(K)}\bigg\}^2\log(KT)/\Delta_*$$
$$+\sqrt{C\bigg\{\frac{K\log(T)}{\beta_1}\left(\frac{\log(T)}{\delta\lambda_{\min}\sqrt{\log(K)}}+d\right)+\beta_1\sqrt{\log(K)}\bigg\}^2\log(KT)/\Delta_*}\Bigg);$$

*that is, from $\beta_1 = C_\Theta C_{\mathcal{X}}^2\sqrt{K\log(T)\left(\frac{\log(T)}{\delta\lambda_{\min}\sqrt{\log(K)}}+d\right)}$,*

$$R_T = O\Bigg(\frac{K\log(K)\log(T)\left(\frac{\log(T)}{\delta\lambda_{\min}\sqrt{\log(K)}}+d\right)\log(KT)}{\Delta_*}+\sqrt{\frac{CK\log(K)\log(T)\left(\frac{\log(T)}{\delta\lambda_{\min}\sqrt{\log(K)}}+d\right)\log(KT)}{\Delta_*}}\Bigg)$$

*holds.*

The result in Corollary 4.3 implies $R_T = O\left(\frac{\log^3(T)}{\Delta_*}+\sqrt{\frac{C\log^3(T)}{\Delta_*}}\right)$ regarding $T$.

*Proof.* From the definition of the contextual adversarial regime with a self-bounding constraint, we have

$$R_T \geq \Delta_* \cdot \mathbb{E}\left[\sum_{t=1}^{T}\Big(1-\pi_t(a^*(X_0)\mid X_0)\Big)\right]-C = \Delta_* \cdot \overline{Q}(a^*)-C.$$

Therefore, from Lemma B.7, for any $\lambda > 0$, we have

$$R_T = (1+\lambda)R_T - \lambda R_T$$

$$= (1+\lambda)O\left(c\sqrt{\log(KT)}\sqrt{\sum_{t=1}^{T}\mathbb{E}\left[H(q_t(X_0))\right]}\right)-\lambda R_T$$

$$\leq (1+\lambda)O\left(c\sqrt{\log(KT)}\sqrt{\sum_{t=1}^{T}\mathbb{E}\left[H(q_t(X_0))\right]}\right)-\lambda\Delta_* \cdot \overline{Q}(a^*)+\lambda C,$$

where

$$c = \left(\frac{K\log(T)}{\beta_1}\left(\frac{\log(T)}{\delta\lambda_{\min}\sqrt{\log(K)}}+d\right)+\beta_1\sqrt{\log(K)}\right).$$

Here, as well as the proof of Theorem 4.1, from Lemma B.8, if $Q(a^*\mid x)\leq e$, we have $\sum_{t=1}^{T}H(q_t(x))\leq e\log(KT)$ and otherwise, we have $\sum_{t=1}^{T}H(q_t(x))\leq Q(a^*\mid x)\log(KT)$. Hence, we have $\sum_{t=1}^{T}H(q_t(x))\leq \log(KT)\max\{e, Q(a^*\mid x)\}$. Here, to upper bound $R_T$, it is enough to only consider a case with $Q(a^*\mid x)\geq e$, and we obtain

$$R_T \leq (1+\lambda)O\left(c\sqrt{\log(KT)}\sqrt{\overline{Q}(a^*)\log(KT)}\right)-\lambda\Delta_* \cdot \overline{Q}(a^*)+\lambda C \leq \frac{O\left(\big\{(1+\lambda)c\big\}^2\sqrt{\log(KT)}\right)}{2\lambda\Delta_*}+\lambda\Delta_*.$$

where the second inequality follows from $a\sqrt{b}-\frac{c}{2}b\leq\frac{a^2}{c^2}$ holds for any $a, b, c > 0$. By choosing

$$\lambda = \sqrt{\frac{c^2\log(KT)}{\Delta_*}\Big/\left(\frac{c^2\log(KT)}{\Delta_*}+2C\right)}.$$

Then, we obtain $R_T = O\left(c^2 \log(KT)/\Delta_* + \sqrt{Cc^2 \log(KT)/\Delta_*}\right)$. □

In the Appendix B, we show the proof procedure of Theorem 4.1.

## 5 Conclusion

We developed a BoBW algorithm for linear contextual bandits. Our proposed algorithm is based on the FTRL approach. In our theoretical analysis, we show that the upper bounds of the proposed algorithm are given as $O\left(\min\left\{\frac{D}{\Delta_*} + \sqrt{\frac{CD}{\Delta_*}}, \sqrt{\log(KT)TD}\right\}\right)$, where $D = K\log(T)\left(\log(T) + d\log(K)\right)\log(KT)$. This regret upper bound implies $O\left(\min\left\{\sqrt{\frac{TD}{\Delta_*}}, \sqrt{\log(KT)TD}\right\}\right)$ regret in an adversarial environment and $O\left(\frac{D}{\Delta_*}\right)$ regret in an adversarial environment and $O\left(\frac{D}{\Delta_*}\right)$ regret in a stochastic environment. This result also implies $O\left(\frac{\log^3(T)}{\Delta_*}\right)$ regret in a stochastic regime and $O\left(\sqrt{T}\log^2(T)\right)$ regret in an adversarial regime with respect to $T$.

There are four directions for future work in this study. The first direction is to develop an algorithm that does not require a contextual distribution while maintaining the BoBW property. We expect this extension can be accomplished by applying our proposed method to a method proposed by Liu et al. (2023), based on the FTRL approach with the log-determinant barrier. We note that standard linear contextual bandits in a stochastic environment do not require the contextual distribution to be known, but it is required for dealing with an adversarial environment.

The second direction is to provide lower bounds in our adversarial regimes. In existing studies, Li et al. (2021) provides a general upper bound that holds for a high-dimensional setting with various margin conditions. We can incorporate such results to derive a lower bound in our problem setting.

The third extension is to develop an algorithm that works for linear contextual bandits without assuming a specific minimum gap constant $\Delta_*$. To address this issue, we might use the margin condition to generalize the minimum gap assumption. Lastly, tightening our regret upper bound is also an open problem.

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

## A  Details of Example 1

When $\min_{b \neq a_0^*(x)} \left\langle x, \theta_0(b) \right\rangle - \left\langle x, \theta_0(a_0^*(x)) \right\rangle \geq \Delta_*$ holds for all $x \in \mathcal{X}$, we have

$$
\begin{aligned}
R_T &= \mathbb{E}\left[ \sum_{t=1}^{T} \ell_t(A_t, X_t) - \sum_{t=1}^{T} \ell_t(a_0^*(X_t), X_t) \right] \\
&= \mathbb{E}\left[ \sum_{t=1}^{T} \sum_{a \in [K]} X_t\Big(\theta_0^*(a) - \theta_0^*(a_0^*(X_t))\Big) \pi_t(a \mid X_t) \right] \\
&= \mathbb{E}\left[ \sum_{t=1}^{T} \sum_{a \neq a_0^*(X_t)} X_t\Big(\theta_0^*(a) - \theta_0^*(a_0^*(X_t))\Big) \pi_t(a \mid X_t) \right] \\
&\geq \Delta_* \cdot \mathbb{E}\left[ \sum_{t=1}^{T} \sum_{a \neq a_0^*(X_t)} \pi_t(a \mid X_t) \right] \\
&\geq \Delta_* \cdot \mathbb{E}\left[ \sum_{t=1}^{T} \Big(1 - \pi_t(a_0^*(X_0) \mid X_0)\Big) \right]
\end{aligned}
$$

## B  Proof of Theorem 4.1

### B.1  Preliminaries for the Proof of Theorem 4.1

Let $X_0$ be a sample from the context distribution $\mathcal{D}$ independent of $\mathcal{F}_T$. Let $D_t(p, q)$ denote the Bregman divergence of $p.q \in \Pi$ with respect to $\psi_t$; that is,

$$
D_t(p, q) = \psi_t(p) - \psi_t(q) - \left\langle \nabla \psi_t(q), p - q \right\rangle.
$$

Let us define $\pi^* \in \Pi$ as $\pi^*(a^*(x) \mid x) = 1$ and $\pi^*(a \mid x) = 0$ for all $a \in [K]\backslash\{a^*(x)\}$.

Then, the following lemma holds. The proof is shown in Appendix C

**Lemma B.1.** *If $A_t$ is chosen as our proposed method, the regret is bounded by*

$$
\begin{aligned}
R_T \leq \mathbb{E}\Bigg[ \sum_{t=1}^{T} \Big\{ &\gamma_t + \left\langle \widehat{\ell}_t(X_0), q_t(X_0) - q_{t+1}(X_0) \right\rangle \\
&- D_t(q_{t+1}(X_0), q_t(X_0)) + \psi_t(q_{t+1}(X_0)) - \psi_{t+1}(q_{t+1}(X_0)) \Big\} \\
&+ \psi_{T+1}(\pi^*(X_0)) - \psi_1(q_1(X_0)) \Bigg] + 2\sum_{t=1}^{T} \max_{a \in [K]} \left| \mathbb{E}\left[ \langle X_t, \theta_t(a) - \widehat{\theta}_t(a) \rangle \right] \right|.
\end{aligned}
$$

To show Lemma B.1, we use the following proposition from Neu & Olkhovskaya (2020).

**Proposition B.2.** *Suppose that $\pi_t \in \mathcal{F}_{t-1}$ and that $\mathbb{E}\left[ \widehat{\theta}_t(a) \mid \mathcal{F}_{t-1} \right] = \theta_t(a)$ for all $t, a$ hold. Then, the following holds:*

$$
\mathbb{E}\left[ \sum_{t=1}^{T} \sum_{a \in [K]} \Big(\pi_t(a \mid X_t) - \pi^*(a \mid X_t)\Big)\left\langle X_t, \theta_t(a) \right\rangle \right] = \mathbb{E}\left[ \sum_{t=1}^{T} \sum_{a \in [K]} \Big(\pi_t(a \mid X_0) - \pi^*(a \mid X_0)\Big)\left\langle X, \widehat{\theta}_t(a) \right\rangle \right].
$$

This proposition plays an important role throughout this study.

**Bounding the stability term.** For the stability term $\left\langle \widehat{\ell}_t(X_0), q_t(X_0) - q_{t+1}(X_0) \right\rangle - D_t(q_{t+1}(X_0), q_t(X_0))$, we use the following proposition from Ito et al. (2022).

**Proposition B.3** (From Lemma 8 in Ito et al. (2022)). *If $\psi_t$ is given as equation 3, for any $\ell : \mathcal{X} \to \mathbb{R}^K$ and $p, q \in \Pi$, we have*

$$\left\langle \ell_t(x), p(x) - q(x) \right\rangle - D_t(q(x), p(x)) \leq \beta_t \sum_{a \in [K]} p(a \mid x) \xi \left( \frac{\ell_t(a, x)}{\beta_t} \right).$$

*for any $x \in \mathcal{X}$, where $\xi(x) := \exp(-x) + x - 1$.*

For $\widehat{\ell}_t(a, x)$, if $\frac{\widehat{\ell}_t(a,x)}{\beta_t} \geq -1$ holds, then Proposition B.3 implies

$$\left\langle \widehat{\ell}_t(x), q_t(x) - q_{t+1}(x) \right\rangle - D_t(q_{t+1}(x), q_t(x)) \leq \frac{1}{\beta_t} \sum_{a \in [K]} \pi_t(a \mid x) \widehat{\ell}_t^2(a, x).$$

For the RHS, we apply the following proposition from Neu & Olkhovskaya (2020).

**Proposition B.4** (From Lemma 6 in Neu & Olkhovskaya (2020)). *For each $t \in [T]$, our strategy satisfies*

$$\mathbb{E} \left[ \sum_{a \in [K]} \pi_t(a \mid X_0) \widehat{\ell}_t^2(a, X_0) \mid \mathcal{F}_{t-1} \right] \leq 3Kd.$$

**Estimation error of the design matrix.** Next, we bound $\sum_{t=1}^T \max_{a \in [K]} \left| \mathbb{E} \left[ \langle X_t, \theta_t(a) - \widehat{\theta}_t(a) \rangle \right] \right|$. An upper bound of $\sum_{t=1}^T \max_{a \in [K]} \left| \mathbb{E} \left[ \langle X_t, \theta_t(a) - \widehat{\theta}_t(a) \rangle \right] \right|$ is given as the following lemma.

**Lemma B.5.** *For all $t$ such that $\gamma_t = \widetilde{\gamma}_t$, we have $\left| \mathbb{E} \left[ \langle X_t, \theta_t(a) - \widehat{\theta}_t(a) \rangle \right] \right| \leq C_{\mathcal{X}} C_{\Theta} / T$.*

*Proof of Lemma B.5.* From Lemma 5 in Neu & Olkhovskaya (2020), we have $\left| \mathbb{E} \left[ \langle X_t, \theta_t(a) - \widehat{\theta}_t(a) \rangle \right] \right| \leq C_{\mathcal{X}} C_{\Theta} \exp \left( -\frac{\gamma_t \delta}{K} \lambda_{\min} M_t \right)$. Then, when $\gamma_t = \widetilde{\gamma}_t$, we have

$$\exp \left( -\frac{\gamma_t \delta}{K} \lambda_{\min} M_t \right) = \exp \left( -\frac{K \log(T)}{\delta \lambda_{\min} \cdot 2\beta_t} \frac{\delta \lambda_{\min}}{K} M_t \right)$$

$$\leq \exp \left( -\frac{K \log(T)}{\delta \lambda_{\min} \cdot (2\beta_t - 1)} \frac{\delta \lambda_{\min}}{K} M_t \right) = \exp \left( -\log(T) \right) = \frac{1}{T},$$

where recall that we defined $M_t = 2\beta_t - 1$. $\qquad \square$

## B.2 Proof of Theorem 4.1

Then, we show the following lemma. The proof is shown in Appendix D.

**Lemma B.6.** *The regret for the* BoBW-RealFTRL *with $\widehat{\Sigma}_{t,a}^{\dagger}$ is bounded as*

$$R_T \leq \mathbb{E} \left[ \sum_{t=1}^T \left\{ \gamma_t + \frac{3Kd}{\beta_t} + (\beta_{t+1} - \beta_t) H(q_{t+1}(X_0)) \right\} \right] + \beta_1 \log(K) + 2C_{\mathcal{X}} C_{\Theta}.$$

From this result, we obtain the following lemma. We provide the proof in Appendix E

**Lemma B.7.** *Assume the conditions in Theorem B.6. Suppose that $\beta_t$ and $\gamma_t$ satisfy equation 4 and equation 5. Then, we have*

$$R_T \leq \bar{c} \sqrt{\mathbb{E} \left[ \sum_{t=1}^T H(q_t(X_0)) \right]} + 2T_0 + 2C_{\mathcal{X}} C_{\Theta},$$

*where*

$$\bar{c} = O\left(\frac{K\log(T)}{\beta_1}\left(\frac{\log(T)}{\delta\lambda_{\min}\sqrt{\log(K)}} + d\right) + \beta_1\sqrt{\log(K)}\right),$$

$$T_0 := \left\lceil \frac{K\log(T)}{2\delta\lambda_{\min}\beta_1\sqrt{\log(K)}}\sqrt{\sum_{s=1}^{T} H(q_s(X_s))} \right\rceil.$$

Next, we consider bounding $\sum_{t=1}^{T} H(q_t(x))$ by $Q(a^* \mid x)$ as shown in the following lemma.

**Lemma B.8** (From Lemma 4 in Ito et al. (2022)). *For any $a^* : \mathcal{X} \to [K]$, the following holds:*

$$\sum_{t=1}^{T} H(q_t(x)) \leq Q(a^* \mid x)\log\left(\frac{eKT}{Q(a^* \mid x)}\right),$$

*where $e$ is Napier's constant.*

By using the above lemmas and propositions, we prove Theorem 4.1.

*Proof of Theorem 4.1.* From Lemma B.8, if $Q(a^* \mid x) \leq e$, we have $\sum_{t=1}^{T} H(q_t(x)) \leq e\log(KT)$ and otherwise, we have $\sum_{t=1}^{T} H(q_t(x)) \leq Q(a^* \mid x)\log(KT)$. Hence, we have $\sum_{t=1}^{T} H(q_t(x)) \leq \log(KT)\max\{e, Q(a^* \mid x)\}$. From Lemma B.7, we have

$$R_T \leq \bar{c}\sqrt{\sum_{t=1}^{T} \mathbb{E}\left[H(q_t(X_0))\right]} + 2C_{\mathcal{X}}C_{\Theta}$$

$$= O\left(\left(\frac{K\log(T)}{\beta_1}\left(\frac{\log(T)}{\delta\lambda_{\min}\sqrt{\log(K)}} + d\right) + \beta_1\sqrt{\log(K)}\right)\sqrt{\log(KT)}\max\left\{\overline{Q}^{1/2}, 1\right\}\right).$$

$\square$

## C   Proof of Lemma B.1

Let us define

$$\widehat{R}_T(x) := \sum_{t=1}^{T}\sum_{a\in[K]}\left(\pi_t(a \mid x) - \pi^*(a \mid x)\right)\left\langle x, \widehat{\theta}_t(a)\right\rangle.$$

Then, the following holds:

$$R_T \leq \mathbb{E}\left[\widehat{R}_T(X_0)\right] + 2\sum_{t=1}^{T}\max_{a\in[K]}\left|\mathbb{E}\left[\langle X_t, \theta_t(a) - \widehat{\theta}_t(a)\rangle\right]\right|.$$

Then, we prove Lemma B.1 as follows.

*Proof of Lemma B.1.* From the definition of our algorithm, we have

$$R_T = \mathbb{E}\left[\sum_{t=1}^{T}\ell_t(A_t, X_t) - \sum_{t=1}^{T}\ell_t(a_T^*, X)\right]$$

$$= \mathbb{E}\left[\sum_{t=1}^{T}\langle\ell_t(X_t), \pi_t(X_t) - \pi^*(X_t)\rangle\right]$$

$$\begin{aligned}
&= \mathbb{E}\left[\sum_{t=1}^{T}\langle \ell_t(X_t), q_t(X_t) - \pi^*(X_t)\rangle + \sum_{t=1}^{T}\gamma_t\left\langle \ell_t(X_t), \frac{1}{K}\iota - q_t(X_t)\right\rangle\right] \\
&\le \mathbb{E}\left[\sum_{t=1}^{T}\left\langle \ell_t(X_t), q_t(X_t) - \pi^*(X_t)\right\rangle + C_\ell\sum_{t=1}^{T}\gamma_t\right] \\
&= \mathbb{E}\left[\sum_{t=1}^{T}\left\langle \ell_t(X_0), q_t(X_0) - \pi^*(X_0)\right\rangle + C_\ell\sum_{t=1}^{T}\gamma_t\right] \\
&= \mathbb{E}\left[\sum_{t=1}^{T}\left\langle \widehat{\ell}_t(X_0), q_t(X_0) - \pi^*(X_0)\right\rangle + C_\ell\sum_{t=1}^{T}\gamma_t\right] + \mathbb{E}\left[\sum_{t=1}^{T}\left\langle \ell_t(X_0) - \widehat{\ell}_t(X_0), q_t(X_0) - \pi^*(X_0)\right\rangle\right] \\
&\le \mathbb{E}\left[\sum_{t=1}^{T}\left\langle \widehat{\ell}_t(X_0), q_t(X_0) - \pi^*(X_0)\right\rangle + C_\ell\sum_{t=1}^{T}\gamma_t\right] + 2\sum_{t=1}^{T}\max_{a\in[K]}\left|\mathbb{E}\left[\left\langle X_0, \theta_t(a) - \widehat{\theta}_t(a)\right\rangle\right]\right|. \quad\quad (6)
\end{aligned}$$

Then, from the definitions of $q_t$, for each $x \in \mathcal{X}$, we also have

$$\begin{aligned}
&\sum_{t=1}^{T}\left\langle \widehat{\ell}_t(x), \pi^*(x)\right\rangle + \psi_{T+1}(\pi^*(x)) \\
&\ge \sum_{t=1}^{T}\left\langle \widehat{\ell}_t(x), q_{T+1}(x)\right\rangle + \psi_{T+1}(q_{T+1}(x)) - \psi_{T+1}(q_{T+1}(x)) \\
&\quad + \psi_{T+1}(\pi^*(x)) - \langle \nabla\psi_t(q_{T+1}(x)), \pi^*(x) - q_{T+1}(x)\rangle \\
&= \sum_{t=1}^{T}\left\langle \widehat{\ell}_t(x), q_{T+1}(x)\right\rangle + \psi_{T+1}(q_{T+1}(x)) + D_{T+1}(\pi^*(x), q_{T+1}(x)),
\end{aligned}$$

where we used that $\langle \nabla\psi_t(q_{T+1}(x)), \pi^*(x) - q_{T+1}(x)\rangle \ge 0$ holds for a convex function $\psi_t$. Then, it holds that

$$\begin{aligned}
&\sum_{t=1}^{T}\left\langle \widehat{\ell}_t(x), \pi^*(x)\right\rangle + \psi_{T+1}(\pi^*(x)) \\
&\ge \sum_{t=1}^{T}\left\langle \widehat{\ell}_t(x), q_{T+1}(x)\right\rangle + D_{T+1}(\pi^*(x), q_{T+1}(x)) + \psi_{T+1}(q_{T+1}(x)) \\
&\ge \sum_{t=1}^{T}\left\langle \widehat{\ell}_t(x), q_{T+1}(x)\right\rangle + \psi_T(q_T(x)) \\
&\quad + D_T(q_{T+1}(x), q_T(x)) + D_{T+1}(\pi^*(x), q_{T+1}(x)) - \psi_T(q_{T+1}(x)) + \psi_{T+1}(q_{T+1}(x)) \\
&= \sum_{t=1}^{T-1}\left\langle \widehat{\ell}_t(x), q_{T+1}(x)\right\rangle + \psi_T(q_T(x)) \\
&\quad + \left\langle \widehat{\ell}_T(x), q_{T+1}(x)\right\rangle + D_T(q_{T+1}(x), q_T(x)) + D_{T+1}(\pi^*(x), q_{T+1}(x)) - \psi_T(q_{T+1}(x)) + \psi_{T+1}(q_{T+1}(x)) \\
&\ge \sum_{t=1}^{T-1}\left\langle \widehat{\ell}_t(x), q_T(x)\right\rangle + \psi_T(q_T(x)) \\
&\quad + \left\langle \widehat{\ell}_T(x), q_{T+1}(x)\right\rangle + D_T(q_{T+1}(x), q_T(x)) + D_{T+1}(\pi^*(x), q_{T+1}(x)) - \psi_T(q_{T+1}(x)) + \psi_{T+1}(q_{T+1}(x)) \\
&\ge \sum_{t=1}^{T}\left\langle \widehat{\ell}_t(x), q_{t+1}(x)\right\rangle + \sum_{t=1}^{T}D_t(q_{t+1}(x), q_t(x)) - \sum_{t=1}^{T}\left\{\psi_t(q_{t+1}(x)) - \psi_{t+1}(q_{t+1}(x))\right\} + \psi_1(q_1(x)).
\end{aligned}$$

Therefore, we have

$$\sum_{t=1}^{T}\left\langle \widehat{\ell}_t(x), q_t(x) - \pi^*(x)\right\rangle$$

$$\leq \sum_{t=1}^{T} \left\{ \left\langle \widehat{\ell}_t(x), q_t(x) - q_{t+1}(x) \right\rangle - D_t(q_{t+1}(x), q_t(x)) + \psi_t(q_{t+1}(x)) - \psi_{t+1}(q_{t+1}(x)) \right\}$$
$$+ \psi_{T+1}(\pi^*(x)) - \psi_1(q_1(x)).$$

Combining this with equation 6, we obtain

$$R_T \leq \mathbb{E} \Bigg[ \sum_{t=1}^{T} \left\{ \left\langle \widehat{\ell}_t(X_0), q_t(X_0) - q_{t+1}(X_0) \right\rangle - D_t(q_{t+1}(x), q_t(X_0)) + \psi_t(q_{t+1}(X_0)) - \psi_{t+1}(q_{t+1}(X_0)) \right\}$$
$$+ \psi_{T+1}(\pi^*(X_0)) - \psi_1(q_1(X_0)) + C_\ell \sum_{t=1}^{T} \gamma_t \Bigg] + 2 \sum_{t=1}^{T} \max_{a \in [K]} \left| \mathbb{E} \left[ \left\langle X_0, \theta_t(a) - \widehat{\theta}_t(a) \right\rangle \right] \right|.$$

$\square$

# D Proof of Lemma B.6

*Proof of Lemma B.6.* From Lemma B.1, we have

$$R_T \leq \mathbb{E} \Bigg[ \sum_{t=1}^{T} \Big( \gamma_t + \left\langle \widehat{\ell}_t(X_0, d), \pi_t(X_0) - q_{t+1}(X_0) \right\rangle - D_t(q_{t+1}(X_0), \pi_t(X_0))$$
$$+ \psi_t(q_{t+1}(X_0)) - \psi_{t+1}(q_{t+1}(X_0)) \Big) + \psi_{T+1}(\pi^*(X_0)) - \psi_1(q_1(x)) \Bigg]$$
$$+ 2 \sum_{t=1}^{T} \max_{a \in [K]} \left| \mathbb{E} \left[ \langle X_t, \theta_t(a) - \widehat{\theta}_t(a) \rangle \right] \right|.$$

First, we show

$$\mathbb{E} \left[ \left\langle \widehat{\ell}_t(X_0), \pi_t(X_0) - q_{t+1}(X_0) \right\rangle - D_t(q_{t+1}(X_0), \pi_t(X_0)) \right] \leq \frac{3Kd}{\beta_t}. \tag{7}$$

To show this, we confirm $\frac{\widehat{\ell}_t(a,x)}{\beta_t} \geq -1$, which is necessary to derive an upper bound from Proposition B.3. We have

$$\frac{1}{\beta_t} \cdot \left\langle X_0, \widehat{\theta}_t(a) \right\rangle = \frac{1}{\beta_t} \cdot X_0^\top \widehat{\Sigma}_{t,a}^\dagger X_t \left\langle X_t, \theta_{t,a} \right\rangle \mathbb{1}[A_t = a] \geq -\frac{C_\ell}{\beta_t} \cdot \left| X_0^\top \widehat{\Sigma}_{t,a}^\dagger X_t \right|$$

$$\geq -\frac{1}{\beta_t} C_\Theta C_{\mathcal{X}}^2 \left\| \widehat{\Sigma}_{t,a}^\dagger \right\|_{\mathrm{op}} \geq -\frac{1}{\beta_t} C_\Theta C_{\mathcal{X}}^2 \delta \left( 1 + \sum_{k=1}^{M_t} \| V_{k,a} \|_{\mathrm{op}} \right) = -\frac{1}{2\beta_t}(M_t + 1),$$

where we used that $\delta = \frac{1}{2C_\Theta C_{\mathcal{X}}^2}$. Here, recall that we defined $M_t$ as $2\beta_t - 1$. Therefore, $\frac{\widehat{\ell}_t(a,x)}{\beta_t} = -1$ holds. Then, we have

$$\left\langle \widehat{\ell}_t(x), \pi_t(x) - q_{t+1}(x) \right\rangle - D_t(q_{t+1}(x), \pi_t(x))$$
$$\leq \beta_t \sum_{a \in [K]} \pi_t(a \mid x) \xi \left( \frac{\widehat{\ell}_t(a,x)}{\beta_t} \right) \leq \frac{1}{\beta_t} \sum_{a \in [K]} \pi_t(a \mid x) \widehat{\ell}_t^2(a,x).$$

Then, from Proposition B.4, we have equation 7.

From $\psi_t(q(x)) = -\beta_t H(q(x))$, we have

$$\sum_{t=1}^{T} \left( \psi_t(q_{t+1}(x)) - \psi_{t+1}(q_{t+1}(x)) \right) + \psi_{T+1}(\pi^*(x)) - \psi_1(q_1(x))$$

$$\leq \sum_{t=1}^{T} (\beta_{t+1} - \beta_t) H(q_{t+1}(x)) + \beta_1 \log(K).$$

From $\beta_{t+1} = \beta_t + \dfrac{\beta_1}{\sqrt{1 + \left(\log(K)\right)^{-1} \sum_{s=1}^{t} H\left(q_s(X_s)\right)}}$, we obtain

$$\beta_t = \beta_1 + \sum_{u=1}^{t-1} \frac{\beta_1}{\sqrt{1 + \left(\log(K)\right)^{-1} \sum_{s=1}^{u} H\left(q_s(X_s)\right)}} \geq \frac{t\beta_1}{\sqrt{1 + \left(\log(K)\right)^{-1} \sum_{s=1}^{t} H\left(q_s(X_s)\right)}}.$$

Here, we have

$$\widetilde{\gamma}_t = \frac{K}{2\delta\lambda_{\min}\beta_t} \log(T) \leq \frac{K}{2\delta\lambda_{\min}t\beta_1} \log(T) \sqrt{1 + \left(\log(K)\right)^{-1} \sum_{s=1}^{t} H\left(q_s(X_s)\right)}$$

$$\leq \frac{K}{2\delta\lambda_{\min}t\beta_1\sqrt{\log(K)}} \log(T) \sqrt{\sum_{s=1}^{T} H\left(q_s(X_s)\right)}.$$

Therefore, $\widetilde{\gamma}_t \leq 1$ holds for all $t \geq T_0$ such that

$$T_0 = \left\lceil \frac{K\log(T)}{2\delta\lambda_{\min}\beta_1\sqrt{\log(K)}} \sqrt{\sum_{s=1}^{T} H\left(q_s(X_s)\right)} \right\rceil.$$

From Lemma B.5, if there exists $T_0 \in \mathbb{N}$ such that $\gamma_t = \widetilde{\gamma}_t$ holds for all $t \geq T_0$, we have

$$\sum_{t=t=1}^{T} \max_{a \in [K]} \left| \mathbb{E}\left[ \langle X_t, \theta_t(a) - \widehat{\theta}_t(a) \rangle \right] \right| \leq T_0 + \sum_{t=1}^{T} C_{\mathcal{X}} C_{\Theta} \frac{1}{T} = 2T_0 + C_{\mathcal{X}} C_{\Theta},$$

where we used $\langle X_t, \theta_t(a) - \widehat{\theta}_t(a) \rangle \leq 2$ □

## E  Proof of Lemma B.7

*Proof of Lemma B.7.* Firstly, we note that the following equality holds:

$$\mathbb{E}\left[ \sum_{t=1}^{T} (\beta_{t+1} - \beta_t) H(q_{t+1}(X_{t+1})) \right]$$

$$= \mathbb{E}\left[ \sum_{t=1}^{T} (\beta_{t+1} - \beta_t) \mathbb{E}\left[ H(q_{t+1}(X_{t+1})) \mid \mathcal{F}_t \right] \right]$$

$$= \mathbb{E}\left[ \sum_{t=1}^{T} (\beta_{t+1} - \beta_t) \mathbb{E}\left[ H(q_{t+1}(X_0)) \mid \mathcal{F}_t \right] \right]$$

$$= \mathbb{E}\left[ \sum_{t=1}^{T} (\beta_{t+1} - \beta_t) H(q_{t+1}(X_0)) \right]$$

We show the following two inequalities:

$$\sum_{t=1}^{T} \left( \gamma_t + \frac{3Kd}{\beta_t} \right) = O\left( \frac{K\log(T)}{\beta_1} \left( \frac{\log(T)}{\delta\lambda_{\min}\sqrt{\log(K)}} + d \right) \sqrt{\sum_{t=1}^{T} H\left(q_t(X_t)\right)} \right) \tag{8}$$

$$\sum_{t=1}^{T} (\beta_{t+1} - \beta_t) H(q_{t+1}(X_{t+1})) = O\left(\beta_1 \sqrt{\log(K)} \sqrt{\sum_{t=1}^{T} H(q_t(X_t))}\right). \tag{9}$$

First, we show equation 8. From $\gamma_t = \min\left\{1, \frac{K}{4\delta\lambda_{\min}\beta_t} \log(T)\right\}$, we obtain

$$\sum_{t=1}^{T} \left(\gamma_t + \frac{3Kd}{\beta_t}\right) \le \sum_{t=1}^{T} \left(\frac{K}{2\delta\lambda_{\min}\beta_t} \log(T) + \frac{3Kd}{\beta_t}\right) = \left(\frac{K}{2\frac{1}{2C_\Theta C_\mathcal{X}^2}\delta\lambda_{\min}} \log(T) + 3Kd\right) \sum_{t=1}^{T} \frac{1}{\beta_t}.$$

From $\beta_{t+1} = \beta_t + \frac{\beta_1}{\sqrt{1 + \left(\log(K)\right)^{-1} \sum_{s=1}^{t} H\left(q_s(X_s)\right)}}$, we obtain

$$\beta_t = \beta_1 + \sum_{u=1}^{t-1} \frac{\beta_1}{\sqrt{1 + \left(\log(K)\right)^{-1} \sum_{s=1}^{u} H\left(q_s(X_s)\right)}} \ge \frac{t\beta_1}{\sqrt{1 + \left(\log(K)\right)^{-1} \sum_{s=1}^{t} H\left(q_s(X_s)\right)}}.$$

Therefore, we have

$$\sum_{t=1}^{T} \frac{1}{\beta_t} \le \sum_{t=1}^{T} \frac{\sqrt{1 + \left(\log(K)\right)^{-1} \sum_{s=1}^{t} H\left(q_s(X_s)\right)}}{t\beta_1} \le \frac{1 + \log(T)}{\beta_1} \sqrt{1 + \left(\log(K)\right)^{-1} \sum_{s=1}^{T} H\left(q_s(X_s)\right)}.$$

By using $H\left(q_1(x)\right) = \log(K)$, we obtain

$$\sum_{t=1}^{T} \left(\gamma_t + \frac{3Kd}{\beta_t}\right) = O\left(\frac{K\log(T)}{\beta_1} \left(\frac{\log(T)}{\delta\lambda_{\min}\sqrt{\log(K)}} + d\right) \sqrt{\sum_{t=1}^{T} H\left(q_{t+1}(X_s)\right)}\right).$$

Next, we show equation 9. From the definitions of $\beta_t$ and $\gamma_t$, we have

$$\sum_{t=1}^{T} (\beta_{t+1} - \beta_t) H(q_{t+1}(X_{t+1})) = \sum_{t=1}^{T} \frac{\beta_1}{\sqrt{1 + \left(\log(K)\right)^{-1} \sum_{s=1}^{t} H\left(q_s(X_s)\right)}} H(q_{t+1}(X_{t+1}))$$

$$= 2\beta_1 \sqrt{\log(K)} \sum_{t=1}^{T} \frac{H\left(q_{t+1}(X_{t+1})\right)}{\sqrt{\log(K) + \sum_{s=1}^{t} H\left(q_s(X_s)\right)} + \sqrt{\log(K) + \sum_{s=1}^{t} H\left(q_s(X_s)\right)}}$$

$$\le 2\beta_1 \sqrt{\log(K)} \sum_{t=1}^{T} \frac{H\left(q_{t+1}(X_{t+1})\right)}{\sqrt{\log(K) + \sum_{s=1}^{t+1} H\left(q_s(X_s)\right)} + \sqrt{\log(K) + \sum_{s=1}^{t} H\left(q_s(X_s)\right)}}$$

$$\le 2\beta_1 \sqrt{\log(K)} \sum_{t=1}^{T} \frac{H\left(q_{t+1}(X_{t+1})\right)}{\sqrt{\sum_{s=1}^{t+1} H\left(q_s(X_s)\right)} + \sqrt{\sum_{s=1}^{t} H\left(q_s(X_s)\right)}}$$

$$= 2\beta_1 \sqrt{\log(K)} \sum_{t=1}^{T} \frac{H(q_{t+1}(X_{t+1}))}{H(q_{t+1}(X_{t+1}))} \left\{\sqrt{\sum_{s=1}^{t+1} H\left(q_s(X_s)\right)} - \sqrt{\sum_{s=1}^{t} H\left(q_s(X_s)\right)}\right\}$$

$$= 2\beta_1 \sqrt{\log(K)} \sum_{t=1}^{T} \left\{\sqrt{\sum_{s=1}^{t+1} H\left(q_s(X_s)\right)} - \sqrt{\sum_{s=1}^{t} H\left(q_s(X_s)\right)}\right\}$$

$$= 2\beta_1 \sqrt{\log(K)} \left\{\sqrt{\sum_{s=1}^{T+1} H\left(q_s(X_s)\right)} - \sqrt{H\left(q_1(X_1)\right)}\right\}$$

$$\leq 2\beta_1\sqrt{\log(K)}\sqrt{\sum_{s=1}^{T}H\big(q_s(X_s)\big)},$$

where we used $\sqrt{H(q_{T+1}(X_{T+1}))} \leq \sqrt{H(q_1(X_1))}$.

From $\beta_{t+1} = \beta_t + \dfrac{\beta_1}{\sqrt{1+\big(\log(K)\big)^{-1}\sum_{s=1}^{t}H\big(q_s(X_s)\big)}}$, we obtain

$$\beta_t = \beta_1 + \sum_{u=1}^{t-1}\frac{\beta_1}{\sqrt{1+\big(\log(K)\big)^{-1}\sum_{s=1}^{u}H\big(q_s(X_s)\big)}} \geq \frac{t\beta_1}{\sqrt{1+\big(\log(K)\big)^{-1}\sum_{s=1}^{t}H\big(q_s(X_s)\big)}}.$$

Inequalities equation 8 and equation 9 combined with the inequality in Lemma B.6 yield

$$R_T \leq \mathbb{E}\left[\sum_{t=1}^{T}\left\{\gamma_t + \frac{3Kd}{\beta_t} + (\beta_{t+1}-\beta_t)\,H(q_{t+1}(X_0))\right\}\right] + \beta_1\log(K) + 2T_0 + 2C_{\mathcal{X}}C_{\Theta}$$

$$= \mathbb{E}\left[\sum_{t=1}^{T}\left\{\gamma_t + \frac{3Kd}{\beta_t} + (\beta_{t+1}-\beta_t)\,H(q_{t+1}(X_{t+1}))\right\}\right] + \beta_1\log(K) + 2T_0 + 2C_{\mathcal{X}}C_{\Theta}$$

$$= \mathbb{E}\left[\sum_{t=1}^{T}\left\{O\left(\frac{K\log(T)\big(\log(T)+\delta\lambda_{\min}d\big)}{\beta_1\delta\lambda_{\min}\sqrt{\log(K)}}\sqrt{\sum_{s=1}^{T}H\big(q_{t+1}(X_s)\big)}\right) + O\left(\beta_1\sqrt{\log(K)}\sqrt{\sum_{t=1}^{T}H(q_t(X_t))}\right)\right\}\right]$$
$$+ \beta_1\log(K) + 2T_0 + 2C_{\mathcal{X}}C_{\Theta}$$

$$= \sum_{t=1}^{T}\left\{O\left(\frac{K\log(T)\big(\log(T)+\delta\lambda_{\min}d\big)}{\beta_1\delta\lambda_{\min}\sqrt{\log(K)}}\sqrt{\sum_{s=1}^{T}\mathbb{E}\big[H\big(q_{t+1}(X_0)\big)\big]}\right) + O\left(\beta_1\sqrt{\log(K)}\sqrt{\sum_{t=1}^{T}\mathbb{E}\left[H(q_t(X_t))\right]}\right)\right\}$$
$$+ \beta_1\log(K) + 2T_0 + 2C_{\mathcal{X}}C_{\Theta}.$$

Thus, we obtain the regret bound in Lemma B.7. $\hfill\square$

