# OpenReview forum: "Best-of-Both-Worlds Linear Contextual Bandits"
_TMLR — Accepted by TMLR_

### Review · Reviewer_94HA · 2024-02-11

**Summary Of Contributions:**

This paper proposes the first best-of-both-worlds algorithm for K-armed linear contextual bandits. To achieve this, the authors first introduce a setting called linear contextual adversarial regime with a self-bounding constraint, which generalizes previous settings and captures both stochastic and adversarial cases. Based on this setting, the authors combine the RealLinEXP3 algorithm with a novel learning rate update rule to achieve best-of-both-worlds regret.

**Audience:**

Yes

**Broader Impact Concerns:**

I did not find any ethical concerns.

**Claims And Evidence:**

Yes

**Requested Changes:**

Please see the weaknesses.

**Strengths And Weaknesses:**

Strengths:

1. The problem is important and it is first studied by the authors.

2. The algorithm and proof sketch are clear.



Weaknesses:

1. The regret bound for the stochastic setting is not instance-optimal. It is beneficial to mention it in the paper and briefly discuss some instance-optimal algorithms (such as [1] and [2]).

2. The technical contributions of this paper are limited. The algorithm and analysis combine [3] and [4].

3. Knowing the context distribution is a strong assumption, although the authors discuss possible solutions to relax it.

4. Some descriptions in the paper could be refined. For example,

* It is better to add some instance-dependent bounds for stochastic setting in Table 1

 * In the third paragraph of section 1.1, the sentence "Combining them, ...... , without any assumption on the existence of $\Delta_*$" seems repeated compared with previous sentences.

* The citations in Remark 2 seem confusing. The author first shows the corruption In Lykouris &Vassilvtiskii (2018) and Gupta et al. (2019) depends on $A_t$ but then says their corruption is determined irrelevant to $A_t$.


[1] Hao B, Lattimore T, Szepesvari C. Adaptive exploration in linear contextual bandit[C]//International Conference on Artificial Intelligence and Statistics. PMLR, 2020

[2] Tirinzoni A, Pirotta M, Restelli M, et al. An asymptotically optimal primal-dual incremental algorithm for contextual linear bandits[J]. Advances in Neural Information Processing Systems, 2020

[3] Gergely Neu and Julia Olkhovskaya. Efficient and robust algorithms for adversarial linear contextual bandits.
In Conference on Learning Theory (COLT), 2020.

[4] Shinji Ito, Taira Tsuchiya, and Junya Honda. Nearly optimal best-of-both-worlds algorithms for online
learning with feedback graphs. In Advances in Neural Information Processing Systems (NeurIPS), 2022.

---

> ### Author Response · Authors · 2024-04-17
> **Response to Reviewer 94HA**
>
> We appreciate the constructive comments received during the review process. Below are our responses to the queries:
>
> -------
> **Q1**: The regret bound for the stochastic setting is not instance-optimal. It would be beneficial to mention this in the paper and briefly discuss some instance-optimal algorithms (such as [1] and [2]).
>
> **A1**: We thank you for the suggestion. We cited the proposed papers and appropriately discussed them on page 4 of our revised draft.
>
> -------
>
> **Q2**: Knowing the context distribution is a strong assumption, although the authors discuss possible solutions to relax it.
>
> **A2**: In the adversarial bandit setting, it is extremely challenging to relax the assumption that requires the contextual distribution without additional assumptions, computational costs, or modifications to the problem setting. In fact, although [3] has successfully removed this assumption, the regret analysis introduces entirely different bias and bonus terms compared to the regret analysis in this study, making it difficult to apply the analytical method used in this research.
>
> -------
>
> **Q3**: It is better to add some instance-dependent bounds for stochastic setting in Table 1.
>
> **A3**: We appreciate your suggestion. We considered this issue, but due to the setting by Neu & Olkhovskaya (2020) and others being different from the usual linear contextual bandit setting, we found it difficult to make a direct comparison and decided against including it in Table 1. To avoid confusion, we will discuss this issue in more detail in the Related Work section and Remark 1.
>
> -------
>
> **Q4**. The citations in Remark 2 seem confusing. The author first shows the corruption In Lykouris &Vassilvtiskii (2018) and Gupta et al. (2019) depends on $A_t$ but then says their corruption is determined irrelevant to  $A_t$.
>
> **A4**: This was a typo. To be precise, "Note that the adversarial corruption depends on \(A_t\) in Zhao et al. (2021), while in Lykouris & Vassilvtiskii (2018), Gupta et al. (2019), and He et al. (2022), the adversarial corruption is determined to be irrelevant to \(A_t\)." We have made the correction.
>
> [1] Hao B, Lattimore T, Szepesvari C. Adaptive exploration in linear contextual bandit. In: International Conference on Artificial Intelligence and Statistics. PMLR, 2020.
> [2] Tirinzoni A, Pirotta M, Restelli M, et al. An asymptotically optimal primal-dual incremental algorithm for contextual linear bandits. Advances in Neural Information Processing Systems, 2020.
> [3] Haolin Liu, Chen-Yu Wei, Julian Zimmert, "Bypassing the Simulator: Near-Optimal Adversarial Linear Contextual Bandits," NeurIPS 2023.

---

### Review · Reviewer_FccE · 2024-02-13

**Summary Of Contributions:**

This paper studies the Best-of-Both-Worlds (BOBW) setting for contextual bandits for the stochastic and adversarial regimes. They propose a new regime called the linear contextual adversarial regime with a self-bounding constraint that depends on the corruption parameter C and minimum gap $\Delta_*$. Previous works in this setting have studied adversarial corruption without the existence of such a minimum gap $\Delta_*$. They propose an algorithm BOBW-RealFTRL that incorporates the Matrix Geometric Resampling (MGR) to estimate the design matrix for calculating the $\widehat{\theta}_t$ and the entropy adaptive rule for exploration. This algorithm is shown to have a regret of $O(\min[\frac{D}{\Delta\_{\star}}+\sqrt{\frac{C D}{\Delta\_{\star}}}, \sqrt{\log (K T) T D}])$ which results in $O(\frac{D}{\Delta\_{\star}})$ for stochastic setting (C=0), and $O(\sqrt{\log (K T) T D})$ in the adversarial regime. Their main theorem 4.1 combines the proof technique of Neu & Olkhovskaya (2020) and Ito et al. (2022). They conduct no experiments to show the efficacy of their algorithm.

**Audience:**

Yes

**Broader Impact Concerns:**

Not applicable.

**Claims And Evidence:**

Yes

**Requested Changes:**

The paper needs to explain these things in more detail

1) Why the dimension $d$ is subsumed by other factors in $D = K \log (T)(\log (T)+d \log (K)) \log (K T)$. In a similar work of RealLinExp3, there is the dimension $d$ in the regret bound that scales as $O(\log (T) \sqrt{K d T})$.

2) Is there a particular significance as to why the $\Delta_*$ is important and should be considered as opposed to other works in your related area that do not work with this $\Delta_*$? Probably this paper is dealing with problem-dependent bound for the BOBW setting, but this needs to be clarified and stated more clearly.

3) Your algorithm needs to $\lambda_\min$, and this was not specified in Assumption 2.1. This should be clearly specified also in the parameter line of Alg 1. This is very important, in the sense knowledge of the $\lambda_\min$ ensures us to quantify what is the minimum information we can gather from sampling in every informative direction.

4) In section 3 the (sort of) exploration parameter $\psi_t(q(x))$ is introduced without explanation. This is related to the entropy adaptive rule. It requires more discussion as to why the entropy shows up. Also, more discussion is needed as to why this specific nature of \beta_t?

5) The Algorithm 1 needs to be written more clearly. The calling of MGR in Algorithm 2 is not clear. Of course, it is used to estimate theta_t but this needs to be stated more clearly in the algo 1.

6) I haven't looked into the appendix of the proof of Theorem 4.1, but reading the proof overview in section 4.1 it is not clear to me what is the key technical novelty. The key novelty that seems to me is the regret decomposition of Lemma 4.4. After that the individual components of this decomposition seem to be handled by using existing results from Neu & Olkhovskaya (2020) and Ito et al (2022). Therefore to me, it seems that the key technical contribution in the main Theorem 4.1 is marginal and needs to be stated more clearly.

7) I understand that the paper is mainly theoretical in nature. Yet it is possible to show some experiments on simulated settings (similar to https://proceedings.mlr.press/v75/abbasi-yadkori18a/abbasi-yadkori18a.pdf). This would greatly increase the strength of the paper.

**Strengths And Weaknesses:**

Strengths:
1) Studying the BOBW setting under stochastic and adversarial settings with a self-bounding constraint is an important area of study.

2) The algorithm looks sound. Using the Matrix Geometric Resampling (MGR) to estimate the design matrix for calculating the $\widehat{\theta}_t$ is a standard approach used by Neu & Olkhovskaya (2020). Similarly, using the entropy adaptive rule for exploration is a standard approach for the adversarial setting used in Ito et al., 2022. Studying this in the context of BOBW leads to new insight into a unified framework for both settings.

3) They derive the regret bound for the BOBW-RealFTRL algorithm in Corollary 4.2 and 4.3 which prima-facie looks sound, but I have some questions on these (see weakness). They give a proof overview of their main theorem 4.1 which relies on standard techniques from Neu & Olkhovskaya (2020) but is adapted to the setting for the contextual adversarial regime with a self-bounding constraint.

Weakness:
1) The paper needs some re-writing as several definitions, implicit assumptions, and the algorithm parameters are not defined properly (see requested changes).

2) In the regret bound of Theorem 4.1 it is strange that the dimension $d$ is not a significant factor (see requested changes).

3) The significant of $\Delta_*$ is not clear to me.

4) Some parameter choices need to be explicitly stated and discussed in section 4 (see requested changes).

5) The key technical novelty for proving Theorem 4.1 seems to be marginal (see requested changes).

6) No experiments are conducted for testing BOBW-RealFTRL algorithms.

---

> ### Author Response · Authors · 2024-04-17
> **Response to Reviewer FccE**
>
> We thank you for your constructive comments. We will diligently correct typos and other minor errors. Below are our responses to your queries:
>
> ---------
>
> **Q1a**: In the regret bound of Theorem 4.1, it is strange that the dimension $d$ is not a significant factor (see requested changes).
>
> **Q1b**: Why is the dimension $d$ subsumed by other factors in $D = K \log(T)(\log(T) + d \log(K)) \log(KT)$? In a similar work of RealLinExp3, there is the dimension $d$ in the regret bound that scales as $O(\log(T) \sqrt{K d T})$.
>
> **A1**: We agree with the reviewer and also consider the dimension $d$ to be a crucial parameter. Indeed, in Theorem 1 and the other main theoretical results, we explicitly denote all parameters in the regret upper bounds. The inclusion of $d$ in $D$ in several parts such as the Introduction and Table 1 is not because it is subsumed, but rather because the regret upper bound is long and complex, and $D$ is used for abbreviation. While it would make the notation lengthy, outside of Table 1 and the Conclusion, we will notate all parameters, keeping the current notation in Table 1 and the Conclusion.
>
> ---------
>
> **Q2a**: The significance of $\Delta_*$ is not clear to me.
>
> **Q2b**: Is there a particular reason why $\Delta_*$ is important and should be considered, as opposed to other works in your related area that do not consider it? This paper probably deals with a problem-dependent bound for the BOBW setting, but this needs to be clarified and stated more clearly.
>
> **A2**: We thank you for pointing this out. First, in the stochastic environment case, asymptotically optimal (problem-dependent) regrets depend on $\Delta_*$, making it a crucial factor. This is indeed necessary for deriving O(polylog(T))-regret. Another reason is specific to the contextual bandit problem. We may relax it by defining the margin condition, which is important future work. In the revised draft, we clarified this relationship with previous studies in Remark 1.
>
> Yasin Abbasi-yadkori, Dávid Pál, and Csaba Szepesvári. Improved algorithms for linear stochastic bandits. In Advances in Neural Information Processing Systems (NeurIPS), 2011.
>
> ---------
>
> **Q3**: Your algorithm needs $\lambda_{\min}$, and this was not specified in Assumption 2.1. This should be clearly specified also in the parameter line of Algorithm 1. This is very important, as knowledge of $\lambda_{\min}$ allows us to quantify the minimum information we can gather from sampling in every informative direction.
>
> **A3**: We thank you for your comment. Indeed, $\lambda_{\min}$ is an important parameter. In our setting, the context distribution $\mathcal{D}$ is assumed to be known, and implicitly, we assumed $\lambda_{\min}$ to be known as well. In the revised manuscript, we have explicitly stated that $\lambda_{\min}$ is known.
>
> In practice, instead of $\lambda_{\min}$, a small constant $\underline{\lambda}$ such that $\lambda_{\min} \geq \underline{\lambda} > 0$, independent of $T$, could also be used to show the same results, thereby slightly relaxing the assumption. However, the upper bound depends on $\underline{\lambda}$, and it is tightest when $\lambda_{\min} = \underline{\lambda}$. Yet, since we also assume $\mathcal{D}$ to be known, which implicitly means $\lambda_{\min}$ is known, we did not pursue this direction.
>
> ---------
>
> **Q4**: In section 3, the (sort of) exploration parameter $\psi_t(q(x))$ is introduced without explanation. This is related to the entropy adaptive rule. It requires more discussion as to why the entropy shows up. Also, more discussion is needed as to why this specific nature of $\beta_t$?
>
> **A4**: We thank you for your feedback. We have added explanations regarding these in the manuscript.

---

> > ### Author Response · Authors · 2024-04-17
> > **(Cont) Response to Reviewer FccE**
> >
> > ---------
> >
> > **Q5**: I haven't looked into the appendix for the proof of Theorem 4.1, but reading the proof overview in section 4.1, it is not clear to me what the key technical novelty is. The key novelty seems to be the regret decomposition of Lemma 4.4. After that, the individual components of this decomposition seem to be handled using existing results from Neu & Olkhovskaya (2020) and Ito et al. (2022). Therefore, it seems to me that the key technical contribution in the main Theorem 4.1 is marginal and needs to be stated more clearly.
> >
> > **A5**: Our contribution is more about proposing an effective algorithm for the new problem of Best-of-both-worlds than about technical contributions. Indeed, while we leverage results from existing research, there are additional technical innovations involved. That is, it is not possible to achieve the results by directly using the findings of Ito et al., 2022, or Neu & Olkhovskaya, 2020 alone. For example, the analysis of BoBW under the context $X_t$, and the need to adjust parameters like $M_t$ and $\beta_t$ in MGR, are elements not present in Ito et al., 2022, or Neu & Olkhovskaya, 2020. However, overall, our contribution lies not in the proof techniques but in proposing a new problem setting and presenting suitable algorithms and regret upper bounds for it. We have added explanations to these points in Section 1.1.
> >
> > ---------
> >
> > **Q6a**: No experiments are conducted for testing the BOBW-RealFTRL algorithms.
> >
> > **Q6b**: I understand that the paper is mainly theoretical in nature. Yet, it is possible to show some experiments on simulated settings (similar to https://proceedings.mlr.press/v75/abbasi-yadkori18a/abbasi-yadkori18a.pdf). This would greatly increase the strength of the paper.
> >
> > **A6**: We thank you for the suggestion. We have added some preliminary experimental results in Section 5.
> >
> > ---------
> >
> > **Q7**: Algorithm 1 needs to be written more clearly. The calling of MGR in Algorithm 2 is not clear. Of course, it is used to estimate $\theta_t$ but this needs to be stated more clearly in Algorithm 1.
> >
> > **A7**: We have added more detailed descriptions regarding the MGR algorithm in pages 7 and 8.

---

### Review · Reviewer_o8Wq · 2024-03-24

**Summary Of Contributions:**

The paper studies the linear contextual bandit problem in two regimes: one setting (adversarial) with i.i.d. context and loss vector chosen by the adversary and the other setting where the loss vector is only perturbed by the adversary C times, and the margin condition is assumed. This work provides the algorithm that simultaneously achieves nearly optimal regret bound in both regimes. The algorithm is based on the FTRL with entropy regulariser.

**Audience:**

Yes

**Broader Impact Concerns:**

This is a theoretical work and mentioned concerns are not applied.

**Claims And Evidence:**

Yes

**Requested Changes:**

Remarks:
1.	Page 4, procedure in a trial: in 1., should it be that \theta_{t,a} are chosen at each time step, but not just values for each loss? Or is it equal?
2.	Definition 2.4: in what practical application this assumption holds? Can you motivate it by example?
3.	Expectation operator on the top of page 7: Condition on F_{t-1} need to be added.
4.	Page 7, the proof of unbiasedness of estimate of the inverse of \Sigma_{t,a} in the middle of the page: there is unneeded inverse of Sigma in last equality.
5.	Conclusion: discussion of removing of \Delta^*. How would the regret looks like, as some margin for the stochastic case is still needed?

**Strengths And Weaknesses:**

My main concern regarding the setting is the following. The paper introduces the new setting with margin condition and changing loss function. (Definition 2.4, Remark 1).  The statement of Definition 2.4 is not clear to me, as no explanation on what \pi_t are not given. Does it has to hold for any policy?

The proof looks correct to me. Given the novelty of the result, I lean towards recommending accepting the paper.

---

> ### Author Response · Authors · 2024-04-17
> **Response to Reviewer o8Wq**
>
> We appreciate the constructive feedback from the reviewers. Below are our responses to the raised queries:
>
> -------
>
> **Q1**: On page 4, in the procedure of a trial: in step 1, should it be that $\theta_{t,a}$ are chosen at each time step, and not just values for each loss? Or are they equivalent?
>
> **A1**: Thank you for pointing this out. Given that $X_t$ has not yet been generated, it is more appropriate to consider that $\theta_t$ is chosen at each time step. In the revised draft, we have modified the explanation of the loss generation. We reorganized the first part of Section 2 and Section 2.1 from the first submitted draft into Sections 2.1-2.3 in the revised draft, defining the linear contextual bandits in Section 2.1 and detailing the loss generation in Section 2.3.
>
> -------
>
> **Q2**: Definition 2.4: In what practical applications does this assumption hold? Can you motivate it with an example?
>
> **A2**: In the current draft, we discuss two significant examples in Example 1 and Example 2, clarifying that they correspond to $C=0$ and $C=T$ respectively. Specifically, $C=0$ corresponds to a stochastic environment, and $C=T$ to an adversarial environment. Intermediate \(C\) values represent a spectrum between these extremes, with higher $C$ indicating a more adversarial nature. We will provide further clarification on this.
>
> -------
>
> **Q3**: The expectation operator on the top of page 7: Condition on $F_{t-1}$ needs to be added.
>
> **A3**: Thank you for the correction. We have included $F_{t-1}$.
>
> -------
>
> **Q4**: On page 7, in the proof of the unbiasedness of the estimate of the inverse of $\Sigma_{t,a}$ in the middle of the page: there is an unneeded inverse of Sigma in the last equality.
>
> **A4**: We appreciate your attention to detail in identifying this typo. We have corrected the error by removing the unnecessary inverse.
>
> -------
>
> **Q5**: Conclusion: discussion on the removal of $\Delta^*$. How would the regret look, as some margin for the stochastic case is still needed?
>
> **A5**: Thank you for your inquiry. We have elaborated on our discussion regarding the introduction of a margin condition in Remark 1. We are currently exploring extensions in this area in a separate project. However, to maintain anonymity, we prefer not to delve into specifics here. We will communicate this to the Action Editor and intend to provide more details in the revised version, should it be accepted.

---

> > ### Comment · Reviewer_o8Wq · 2024-04-21
> >
> > Thank you for the clarifications!

---

### Review · Reviewer_YTsy · 2024-03-25

**Summary Of Contributions:**

In this work, the authors studied the problem of achieving best-of-both-world guarantees for linear bandits with stochastic context. Concretely, given then context $X_t$, at each round $t$, the loss of action $a\in [K]$ is defined as $X_t^\top\theta_t(a)+\epsilon_t(a)$. The authors consider three cases: (1) \theta_t is stochastic and there is a unique optimal action and a minimum suboptimality gap $\Delta_*$. (2) stochastic \theta_t with corruption $C$; (3) adversarial $\theta_t$. The authors propose an algorithm based on a combination of RealLinExp3 [Neu & Olkhovskaya, 2020] and FTRL with adaptive learning rate [Ito et al., 2022], and achieve $O(\log^3 T/\Delta_*)$ regret in the stochastic case, $O(\sqrt{dKT\log^3(KT)})$ regret in the adversarial case, and corresponding results in the stochastic corrupted case.

**Audience:**

Yes

**Broader Impact Concerns:**

I do not see concerns on the ethical implications of this work.

**Claims And Evidence:**

Yes

**Requested Changes:**

- Questions in "weakness" section.
- Can the authors include more discussions on other related literatures about BOBW for MAB, including using various other regularizers under different assumptions? This may be related to the question on whether the uniqueness of the optimal action can be removed and whether better than $K/\Delta_*$ can be obtained for the stochastic case.
- There are several places that are unclear to me in the writing:
  - In Remark 2, it is unclear whether results in [Lykouris & Vassilvtiskii, 2018], [Gupta et al, 2019] and [He et al., 2022] depend on the actual played action or not. The sentences are contradicting with each other.
  - In Appendix A, $Q_t^2$ is not defined clearly.
  - In the definition of $\Sigma^{\dagger}_{t,a}$, $X_0$ needs to be $X_t$ since $A_t$ is dependent on $X_t$ instead of $X_0$.

**Strengths And Weaknesses:**

Strength:
- The problem of best-of-both-worlds for linear bandits with stochastic context is not studied before and is well motivated.
- I checked the proofs and the proofs in general look correct to me, with some concerns included in the "weakness" and "requested changes".
- The proposed algorithm is intuitive and easy to implement.

Weakness:
- The main issue is the novelty of the proposed algorithm. While this problem is not studied, the proposed algorithm is a combination of RealLinExp3 [Neu & Olkhovskaya, 2020] and FTRL with adaptive learning rate [Ito et al., 2022], which also does not seem to incur much challenge based on my understanding.
- The results for stochastic setting require several assumptions that may be restrictive. First, the algorithm requires the knowledge of the context distribution as well as the minimum eigenvalue that is used in the tuning of the algorithm, which may not be available (this issue also appears in the adversarial setting). Second, while it is not explicitly mentioned in this paper, the algorithm should only work when the optimal action is unique, which is also assumed in [Ito et al., 2022]. There are many follow-up works trying to relax this assumption including [1,2] with different types of regularizers including log-barrier and Tsallis-INF. I wonder whether other regularizers can be incorporated into RealLinExp3 and combining with these algorithms may lead to better regret bounds with milder assumptions?
- The regret rate obtained may not be ideal, especially for the stochastic environment. Specifically, the rate depends on the minimum suboptimality gap and there is cubic \log T dependency, which are both important for stochastic environment.
- There are some places that are unclear in the writing of the paper in the current version. See "Requested Changes" for detail.


[1] Jin et al., Improved Best-of-Both-Worlds Guarantees for Multi-Armed Bandits: FTRL with General Regularizers and Multiple Optimal Arms, NeurIPS 2023

[2] Shinji Ito, Parameter-Free Multi-Armed Bandit Algorithms with Hybrid Data-Dependent Regret Bounds, COLT 2021

---

> ### Author Response · Authors · 2024-04-17
> **Response to Reviewer YTsy**
>
> Thank you for your constructive comments. We will diligently correct typos and other minor errors. Below are our responses to your queries:
>
> ----------
>
> **Q1**: While this problem has not been studied, the proposed algorithm combines RealLinExp3 [Neu & Olkhovskaya, 2020] and FTRL with an adaptive learning rate [Ito et al., 2022], which, based on my understanding, does not seem to pose much challenge.
>
> **A1**: The contribution of this paper lies more in proposing an effective algorithm for the new problem of the best of both worlds than in technical contributions. Indeed, while we leverage results from existing research, additional technical innovations are involved. That is, it is not possible to achieve the results solely by directly using the findings of Ito et al., 2022, or Neu & Olkhovskaya, 2020 alone. For example, we need to tune the parameters in the inverse covariance matrix estimation (MGR algorithm) as well as the adaptive parameters in regularization. This is because, to obtain an ideal regret, we balance the estimation error of the inverse covariance matrix and regret. Since adaptive regularization parameters do not appear in Neu & Olkhovskaya, 2020, and inverse covariance estimation does not appear in Ito et al., 2022, it has been unclear how we solve the problem. We clarified novelties about such techniques in Section 1.1.
>
> ----------
>
> **Q2**: The results for the stochastic setting require several assumptions that may be restrictive. First, the algorithm requires knowledge of the context distribution as well as the minimum eigenvalue that is used in tuning the algorithm, which may not be available (this issue also appears in the adversarial setting). Second, while it is not explicitly mentioned in this paper, the algorithm should only work when the optimal action is unique, which is also assumed in [Ito et al., 2022]. Many follow-up works are trying to relax this assumption, including [1,2] with different types of regularizers including log-barrier and Tsallis-INF. I wonder whether other regularizers can be incorporated into RealLinExp3 and combined with these algorithms to lead to better regret bounds with milder assumptions?
>
> **A2**: To address the issue, extending to algorithms based on log-barrier or Tsallis-INF is promising, but there are several difficulties that have not been addressed in the bandit problem without contextual information.
> First, regarding the relaxation of the knowledge of the context distribution, even without considering the best of both worlds, only a few studies have succeeded in it, such as Liu et al. (2023). Additionally, even Liu et al. (2023) performs theoretical analysis with bias and bonus terms that are completely different from ours; therefore, applying their results straightforwardly seems to be intractable.
> Second, regarding the relaxation of the unique optimal arms, although it may be possible to introduce another regularization or tune the parameters, the algorithm and analysis will become significantly complicated. Until now, the existing studies have succeeded only for multi-armed bandits without contexts. Even under such a simple setting, the proofs extend over tens of pages. Additionally, there has not yet been an extension to combinatorial semi-bandits or linear bandits, which are considered simpler than contextual linear bandits. Therefore, it seems there are still significant hurdles to removing the assumption of unique optimality in contextual linear bandits.
>
> Haolin Liu, Chen-Yu Wei, and Julian Zimmert. Bypassing the simulator: Near-optimal adversarial linear contextual bandits. In Advances in Neural Information Processing Systems (NeurIPS), 2023.
>
> ----------
>
> **Q3**: The regret rate obtained may not be ideal, especially for the stochastic environment. Specifically, the rate depends on the minimum suboptimality gap, and there is a cubic $\log T$ dependency, which is important for the stochastic environment.
>
> **A3**: As pointed out by the reviewer, our results do not achieve the $O(\log T)$ bound, which presents an opportunity for improvement. For instance, employing the Tsallis-INF method could be considered [citation]. However, using the Tsallis entropy might necessitate more complex computations and assumptions, and even if it achieves a $O(\log T)$ regret, it does not necessarily indicate its superiority over our algorithm. This point is elaborated in the revised Section 1.2.

---

> ### Author Response · Authors · 2024-04-17
> **Re: Response to Reviewer YTsy**
>
> **Q4**: Can the authors include more discussions on other related literature about BOBW for MAB, including using various other regularizers under different assumptions? This may relate to the question of whether the uniqueness of the optimal action can be removed and whether better than $K/\Delta_*$ can be obtained for the stochastic case.
>
> **A4**: Thank you for the suggestion. Following your advice, we have cited a few more papers about other regularizers. We also mention that such an extension is an important future work.
>
> ----------
>
> **Q5**: In Remark 2, it is unclear whether the results in [Lykouris & Vassilvtiskii, 2018], [Gupta et al., 2019], and [He et al., 2022] depend on the actual played action or not. The sentences are contradicting each other.
>
> **A5**: This was a typo on our part. To be precise, "Note that the adversarial corruption depends on $A_t$ in Zhao et al. (2021), while the adversarial corruption is determined to be irrelevant to $A_t$ in Lykouris & Vassilvtiskii (2018), Gupta et al. (2019), and He et al. (2022)." We have made the correction.
>
> ----------
>
> **Q6**: In Appendix A, $Q^2_t$ is not defined clearly.
>
> **A6**: Thank you for pointing this out. We have made the appropriate corrections.
>
> ----------
>
> **Q7**: In the definition of $\Sigma^\dagger_{t, a}$, $X_0$ should be $X_t$ since $A_t$ is dependent on $X_t$ instead of $X_0$.
>
> **A7**: Thank you for pointing out the typo. We have made the correction.

---

### Decision · Action_Editor_R7Lm · 2024-05-11

**Recommendation:** Accept as is

**Comment:**

The reviewers acknowledged the importance of the studied problem, and think the paper introduces novel algorithms and provides theoretical guarantees for the proposed methods. Although using technique similar to [Neu & Olkhovskaya, 2020] and [Ito et al., 2022], the theoretical analysis has some significance. There are some concerns about the algorithm's requirement of knowing the context distribution and a suboptimal regret bound (than O(log T)), but these are understandably challenging to address.

Two requests to the authors for the final version:
- As discussed, can the authors discuss concurrent work https://arxiv.org/abs/2312.15433 in the final version?
- It was mentioned by the authors that some preliminary experimental results are provided in Section 5, but the AE cannot find it. Can the authors include them in the final version?

**Audience:**

Yes

**Claims And Evidence:**

Yes

---

> ### Author Response · Authors · 2024-06-12
> **Reply to AE**
>
> Dear AE,
>
> Thank you for organizing the discussion.
>
> We apologize for the delay in submitting the camera-ready version. Could you please extend the deadline slightly?
>
> First, it was our mistake to mention in our reply that we included the experiments. We initially intended to exclude them. Although we did conduct the experiments, the main contribution of this paper lies in the theoretical aspects rather than the experiments. Additionally, we worried about the reproducibility of the code in the previous rebuttal.
>
> Therefore, may we proceed without including the experiments? We temporally updated the camera-ready version without including the experiments. If it is necessary to include the experiments, please allow us an additional two to three weeks.
>
> Thank you for your understanding and consideration.
>
> Sincerely,
> Authors

---

> > ### Comment · Action_Editor_R7Lm · 2024-06-14
> > **OK to not include experiments**
> >
> > Dear authors,
> >
> > I checked with the reviewers and they are OK with not including experiments. They also suggested that including experiments would have increased the appeal of the paper and show BoBW-RealFTR is easy to implement and is realistic;  this will improve over prior works (Neu & Olkhovskaya, 2020 and Ito et al., 2022), which do not show any experiments even on synthetic setups.
> >
> > So I will leave the decision to you -- If you decide to include the experiments, I can make a request to EIC to extend the deadline for 2-3 weeks. Just let me know!

---

> ### Author Response · Authors · 2024-06-17
> **Thank you for your reply**
>
> Dear AE,
>
> We deeply appreciate your kindness. We intend to submit the current version without the experiments as the final camera-ready version.
>
> However, acknowledging your suggestion that sharing the experimental results would be beneficial to the community, we plan to conduct the experiments independently of the paper. Once completed, we will share the experimental results on platforms like GitHub and notify this OpenReview accordingly.
>
> Thank you for your understanding.
>
> Sincerely,
> Authors